# BREAKING MEMORY AND COMMUNICATION BARRIERS IN MODEL-PARALLEL FINE-TUNING OF LARGE LANGUAGE MODELS

## ABSTRACT

Model parallelism (MP) has emerged as a promising paradigm for distributed large language model (LLM) training across multiple computing nodes. Yet, almost all existing works about MP focus on first-order methods, which faces two persistent challenges: high communication costs from transmitting activations and gradients, and substantial memory overhead from caching them. Zeroth-order (ZO) methods, by avoiding gradient computation and storage, can naturally alleviate both memory and communication bottlenecks, but they have been largely unexplored in MP for LLM fine-tuning. In this work, we propose **SparQ**, a ZO MP framework with **Sp**lit layer **a**llocation info**r**med by **Q**uantization-induced activation sparsity, designed to reduce memory and communication costs. SparQ builds on three key components: (1) leveraging the gradient-free nature of ZO optimization to eliminate gradient storage and transmission, significantly reducing memory and communication demands incurred by gradients; (2) applying quantization to induce activation sparsity that can be encoded with sparse representations; (3) strategically placing split layers at activation-sparse regions and using sparse representation to lower communication cost from activations almost without compromising model quality. Theoretically, SparQ achieves a sublinear convergence rate in non-convex settings, matching that of centralized ZO methods. Empirically, SparQ reduces GPU memory usage by over 3× and communication cost by 50%+ compared to state-of-the-art baselines, while maintaining comparable model performance.

## 1 INTRODUCTION

Large language models (LLMs) have demonstrated strong generalization capabilities, driving their adoption across diverse downstream tasks (Vaswani, 2017; Zhao et al., 2023; Naveed et al., 2025). However, fine-tuning such massive models remains challenging: it requires storing billions of parameters and extra information, such as activations and gradients (Kaddour et al., 2023; Han et al., 2024), often exceeding the memory capacity of a single GPU or machine. Model parallelism (MP), which partitions models across multiple nodes, is a widely used solution, with frameworks such as Megatron-LM (Shoeybi et al., 2019), ZeRO (Rajbhandari et al., 2020), DeepSpeed (Rasley et al., 2020), and MegaScale (Jiang et al., 2024) demonstrating its effectiveness. Most existing works focus on applying first-order (FO) method in the MP scenario. However, the combination of FO and MP inevitably introduce two substantial costs: (1) **High Communication Cost:** FO methods (e.g., SGD) require frequent exchange of large gradients and activations across nodes, making communication a major bottleneck; (2) **High Memory Cost:** FO methods also demand storing gradients, optimizer states, and cached activations, further straining memory and often exceeding a single node's capacity. Scaling typically requires more GPUs and aggressive parallelization, which amplifies both communication and hardware costs.

Recently, zeroth-order (ZO) methods have gained significant attention because they only require forward passes, offering substantial memory savings (Malladi et al., 2023; Zhang et al., 2024; Zhao et al., 2025). However, existing ZO research largely focuses on optimization theory or single-node setups, but the potential of applying ZO optimization within a MP framework for fine-tuning LLMs remains unexplored. This gap naturally raises a key research question:

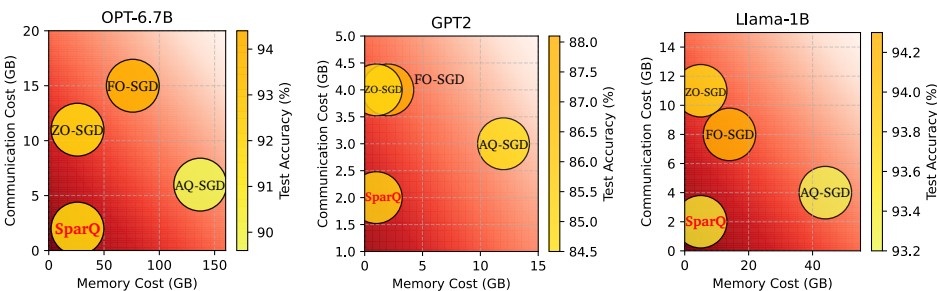

Figure 1: Illustration of Memory Cost, Communication Cost and Test Accuracy. Here we use SST-2 to fine-tune ReLU-based OPT, GELU-based GPT and SwiGLU-based Llama models.

> *Q: What extra, undiscovered advantages in terms of communication and memory cost can be realized by combining ZO and MP for fine-tuning LLMs?*

The major contributions of our work are summarized as follows:

- We empirically observe employing quantization to immediate outputs after commonly used activation functions (e.g., ReLU, GELU, SwiGLU) in Transformer (Vaswani, 2017) architectures during LLM fine-tuning can result in pervasively sparse activations, with the proportion of nonzero entries across layers ranging between $1\% \sim 30\%$.

- Inspired by such inherent sparsity, we propose **`SparQ`**, a ZO model-parallel framework with **Sp**lit layer **a**llocation info**r**med by **Q**uantization-induced activation sparsity. It includes three key components: **(1) ZO optimization for gradient-free fine-tuning**: Prior studies have shown that ZO optimization is effective for LLM fine-tuning due to the presence of low-dimensional subspaces or low-rank Hessian landscapes (Malladi et al., 2023). We exploit its gradient-free nature to eliminate the need for storing and transmitting gradients in MP-based fine-tuning, significantly reducing both memory and communication overhead introduced by gradients. **(2) Quantization-induced high activation sparsity**: In Sec. 2.4, we observe that applying quantization to intermediate outputs after activation functions in Transformer architectures induces higher sparsity than unquantized activations during LLM fine-tuning. **(3) Split layer allocation guided by activation sparsity**: `SparQ` strategically places split layers at these naturally sparse regions. By transmitting activations in sparse representation (i.e., only transmit nonzero entries and their indices) across partitions, communication cost is substantially reduced while maintaining model performance.

- We prove that `SparQ` across multiple nodes achieves a sublinear convergence rate of $\mathcal{O}(\sqrt{d/T})$ for non-convex functions, matching the rate of centralized ZO stochastic gradient descent (ZO-SGD).

- We evaluate `SparQ`'s test accuracy, memory and communication costs on various LLM fine-tuning tasks and observe that: (1) `SparQ` achieves superior efficiency when memory and communication costs are jointly considered while maintaining comparable model performance, as shown in Fig. 1; (2) `SparQ` reduces communication cost by about $50\% \sim 70\%$, compared to FO-SGD and the state-of-the-art activation compression method AQ-SGD (Wang et al., 2022).

## 2 FORMULATIONS AND PRELIMINARIES

Given the loss function $f$, our goal is to minimize the following objective function

$$\min_{\boldsymbol{x} \in \mathbb{R}^d} f(\boldsymbol{x}) := \mathbb{E}_{\xi \sim \mathcal{D}}\left[f(\xi; \boldsymbol{x})\right],$$

where $\boldsymbol{x}$ is a $d$-dimensional model parameter, and $\xi$ is a data sample (model input) selected from the distribution $\mathcal{D}$. In the following subsections, we formulate two core components of `SparQ`: 1) model parallelism technique, and 2) ZO optimization's gradient estimation methods.

### 2.1 MODEL PARALLELISM FORMULATION

We begin by formulating model parallelism. In a model-parallel setting, the entire model is partitioned into $M$ parts, typically, with each computing node responsible for a distinct subset of the model's parameters. For clarity and brevity, we focus on the scenario in our main paper where the model is only divided into two submodels ($M = 2$), each residing on a separate node. This facilitates a

concise presentation of the mathematical formulations, and an extension to an arbitrary number of partitions ($M > 2$) is provided in Appendix C.3.

The partition of model parameters denotes as $\boldsymbol{x} = \mathrm{col}[\boldsymbol{x}^{(1)}, \boldsymbol{x}^{(2)}]$, where $\boldsymbol{x} \in \mathbb{R}^d$, $\boldsymbol{x}^{(1)} \in \mathbb{R}^{d_1}$, $\boldsymbol{x}^{(2)} \in \mathbb{R}^{d_2}$, and $d_1 + d_2 = d$. Under this configuration, the loss function can be written as the composition of two functions:

$$f(\xi; \boldsymbol{x}) := F\left(S_2\big(S_1(\xi; \boldsymbol{x}^{(1)}); \boldsymbol{x}^{(2)}\big)\right), \tag{1}$$

where $F$ is the criterion function, $S_i$ represents a subset of network layers located on node $\mathcal{M}_i$ with model parameters $\boldsymbol{x}^{(i)}$. Specifically, for a given input $\xi$, $S_i(\xi; \boldsymbol{x}^{(i)})$ computes the forward activation on node $\mathcal{M}_i$, which is then transferred to the next node $\mathcal{M}_{i+1}$ where $S_2(\cdot; \boldsymbol{x}^{(2)})$ continues the computation. For notational simplicity, we also express the loss function using operator notation:

$$f(\xi; \boldsymbol{x}) = \left(F \circ S_2|_{\boldsymbol{x}^{(2)}} \circ S_1|_{\boldsymbol{x}^{(1)}}\right)(\xi), \tag{2}$$

where $\circ$ denotes function composition (applying one function to the result of another). $S_1|_{\boldsymbol{x}^{(1)}}$ represents the first sub-model parameterized by $\boldsymbol{x}^{(1)}$ and acting on $\xi$. $S_2|_{\boldsymbol{x}^{(2)}}$ represents the second sub-model parameterized by $\boldsymbol{x}^{(2)}$ and acting on the output of $S_1|_{\boldsymbol{x}^{(1)}}$.

This two-node case ($M = 2$) can naturally be generalized to the multi-node case ($M > 2$) by further partitioning the model. Accordingly, the loss functions in (1) and (2) become

$$f(\xi; \boldsymbol{x}) = F\Big(S_M\big(\cdots\big(S_2\big(S_1(\xi; \boldsymbol{x}^{(1)}); \boldsymbol{x}^{(2)}\big); \cdots\big); \boldsymbol{x}^{(M)}\big)\Big) = \big(F \circ S_M|_{\boldsymbol{x}^{(M)}} \circ \cdots \circ S_2|_{\boldsymbol{x}^{(2)}} \circ S_1|_{\boldsymbol{x}^{(1)}}\big)(\xi).$$

This formulation clearly delineates the flow of data across nodes in a MP framework, laying the groundwork for our subsequent discussions on optimization, memory and communication efficiency.

## 2.2 ZEROTH-ORDER OPTIMIZATION FORMULATION

Next, let us recap the fundamentals of ZO optimization. It is a gradient-free method that approximates the gradient by utilizing the finite difference method rather than explicit differentiation. Given a smoothing parameter $\mu > 0$, the number of perturbations $P$ and random perturbation vectors $\boldsymbol{u}$ drawn from a Gaussian distribution or a uniform ball, the ZO gradient estimate $\hat{G}$ can be computed as:

$$\hat{G} = \frac{1}{P} \sum_{i=1}^{P} g_i \cdot \boldsymbol{u}_i = \frac{1}{P} \sum_{i=1}^{P} \frac{f(\xi; \boldsymbol{x} + \mu\boldsymbol{u}_i) - f(\xi; \boldsymbol{x})}{\mu} \cdot \boldsymbol{u}_i, \tag{3}$$

where Eq. (3) represents a biased forward difference approach. We distribute the content about unbiased central difference approach to Sec. C.2. These two approaches have been widely used to estimate the gradient in a ZO context (Ghadimi & Lan, 2013; Liu et al., 2020).

## 2.3 MODEL PARALLELISM MEETS ZEROTH-ORDER OPTIMIZATION

Having established the formulation for both ZO optimization and model parallelism, we now describe how to integrate ZO optimization into a model-parallel framework. For clarity, we still focus on the two-node case ($M = 2$) here and distributed the multi-node case ($M > 2$) to Appendix C.3. In this setup, node $\mathcal{M}_1$ is responsible solely for computing and transmitting the forward activations, while node $\mathcal{M}_2$ performs the gradient estimation, parameter updates and gradient scalar transmission.

In this work, we utilize classic Zeroth-Order Stochastic Gradient Descent (ZO-SGD) as our optimizer. Specifically, within the second submodel on $\mathcal{M}_2$, the ZO gradient scalar $g_t$ is computed using a forward difference method as follows:

$$g_{i,t} = \frac{F(a_{i,t}^+; \boldsymbol{x}_t^{(2)} + \mu\boldsymbol{u}_{i,t}^{(2)}) - F(a_{i,t}; \boldsymbol{x}_t^{(2)})}{\mu}, \tag{4}$$

where $t$ is the iteration index, and $\boldsymbol{u}_{i,t}^{(2)}$ is the perturbation vector generated on node $\mathcal{M}_2$ during the $t$-th iteration. $a^+$ and $a$ are the activations obtained from the preceding sub-model on $\mathcal{M}_1$, defined as

$$a_{i,t}^+ = S_1\big(\xi_t; \boldsymbol{x}_t^{(1)} + \mu\boldsymbol{u}_{i,t}^{(1)}\big), \quad a_{i,t} = S_1\big(\xi_t; \boldsymbol{x}_t^{(1)}\big). \tag{5}$$

Then, the final zeroth-order gradient estimate is given by $\hat{G}_t = \frac{1}{P} \sum_{i=1}^{P} g_{i,t} \cdot \boldsymbol{u}_{i,t}^{(2)}$.

Integrating ZO optimization into a model-parallel framework offers a promising avenue to alleviate memory burdens associated with large-scale training. Unlike traditional FO methods that require storing gradients, ZO optimization relies exclusively on forward passes, thereby eliminating memory

usage from storing gradients. However, this integration introduces a unique communication challenge. Since ZO optimization often depends on multiple perturbations to approximate gradients accurately, each computing node must transmit several forward activations across split layer per iteration. In contrast, FO methods generally exchange only a single forward activation and one backward gradient per iteration. Thus, while ZO optimization enables significant memory savings, its deployment in MP necessitates careful optimization of communication overhead to maintain training stability and performance. In Sec. 3, we will introduce how we address communication bottleneck.

## 2.4 KEY OBSERVATIONS

We first summarize several key observations from Fig. 2 that motivate SparQ.

**(1) High Sparsity of Original ReLU-Based Activations in LLM Fine-Tuning.** Li et al. (2023) reported that ReLU-based activations exhibit high sparsity during LLM pre-training. Consistent with this finding, we observe a similar pattern in LLM fine-tuning in Fig. 2a: the proportion of nonzero values in nearly all ReLU-based activations remains below $10\%$. **(2) Low Sparsity of Original SwiGLU- and GELU-based Activations in LLM Fine-Tuning.** Smoother activation functions, such as SwiGLU and GELU, produce predominantly small but nonzero values, resulting in consistently dense activations with $99\% \sim 100\%$ nonzero entries. **(3) Quantization Induces High Sparsity across Various Activations.** Applying quantization can dramatically enhance activation sparsity during fine-tuning. Specifically, by using 4-bit quantization, ReLU-based activations become even sparser ( below $5\%$ nonzero entries), SwiGLU-based activations stabilize below $20\%$, and nearly all GELU-based activations fall under $5\%$ nonzero entries. This high sparsity motivates us to split the model immediately after the activation functions since activations can be efficiently encoded in sparse representation (described in Sec. C.1), significantly reducing the communication burden associated with their transmission because we only need to transmit the nonzero values and their indices. The specific split layer selections in our experiments are described in Sec. B.1.

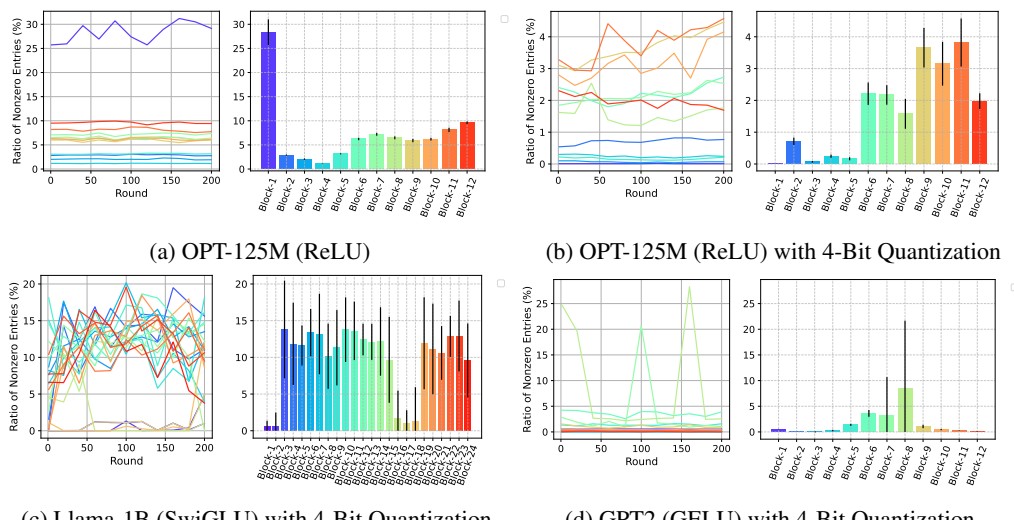

(a) OPT-125M (ReLU)        (b) OPT-125M (ReLU) with 4-Bit Quantization

(c) Llama-1B (SwiGLU) with 4-Bit Quantization  (d) GPT2 (GELU) with 4-Bit Quantization

Figure 2: Sparsity Levels of Various Activations Here we do not show the sparsity level of SwiGLU and GELU because they are very dense. Specifically, their ratio of nonzero entries are 99%-100%.

## 3 SPARQ FRAMEWORK DESIGN

We introduce **SparQ**, a zeroth-order model-parallel framework with with **Sp**lit layer **a**llocation info**r**med by **Q**uantization-induced activation sparsity, reducing communication cost and minimizing memory costs for LLM fine-tuning in model parallelism. It employs ZO optimization to bypass the need for storing and transmitting backward gradients. It is crucial to highlight our split strategy that is closely related to communication reduction. By capitalizing on the quantization-induced high sparsity of activations immediately after activation functions, we partition the model at these sparse areas to naturally cut down on communication costs.

In Alg. 1, we illustrate how SparQ operates on a two-GPU setup ($M = 2$) for brevity and distribute the multi-node case ($M > 2$) to Appendix C.3 due to the limited space. The entire model is partitioned

---

**Algorithm 1** SparQ with Forward Difference Method ($M = 2$)

---

1: **Initialize**: split model immediately after activation functions to get submodels $S_1$ and $S_2$, model parameter $\boldsymbol{x}_0 = \text{col}[\boldsymbol{x}_0^{(1)}, \boldsymbol{x}_0^{(2)}]$, learning rate $\eta$, smoothing parameter $\mu$, iterations $T$.
2: **for** $t = 0, 1, \cdots, T - 1$ **do**
3:     On node $\mathcal{M}_1$:
4:         Sample a random seed $s$ and a data sample $\xi_t$.
5:         Compute $a_t = S_1\big(\xi_t\,;\,\boldsymbol{x}_t^{(1)}\big)$
6:         $\boldsymbol{x}_t^{(1)} \leftarrow \text{Perturb}(\boldsymbol{x}_t^{(1)}, \mu, s)$
7:         Compute $a_t^+ = S_1\big(\xi_t\,;\,\boldsymbol{x}_t^{(1)}\big)$
8:         $\boldsymbol{x}_t^{(1)} \leftarrow \text{Perturb}(\boldsymbol{x}_t^{(1)}, -\mu, s)$
9:         Send $\mathbb{C}(a_t^+)$ and $\mathbb{C}(a_t)$, which are sparse representations of quantized $a_t^+$ and $a_t$, to $\mathcal{M}_2$.
10:    On node $\mathcal{M}_2$:
11:       Compute $f_t = F\big(S_2(\mathbb{C}(a_t)\,;\,\boldsymbol{x}_t^{(2)})\big)$
12:       $\boldsymbol{x}_t^{(2)} \leftarrow \text{Perturb}(\boldsymbol{x}_t^{(2)}, \mu, s)$
13:       Compute $f_t^+ = F\big(S_2(\mathbb{C}(a_t^+)\,;\,\boldsymbol{x}_t^{(2)})\big)$
14:       $\boldsymbol{x}_t^{(2)} \leftarrow \text{Perturb}(\boldsymbol{x}_t^{(2)}, -\mu, s)$
15:       $\hat{g}_t = \frac{1}{\mu}(f_t^+ - f_t)$
16:       $\boldsymbol{x}_{t+1}^{(2)} \leftarrow \text{Update}(\boldsymbol{x}_t^{(2)}, \eta, \hat{g}_t, s)$
17:       Send $\hat{g}_t$ back to $\mathcal{M}_1$.        ▶ Only a scalar is transmitted in backward propagation
18:    On node $\mathcal{M}_1$:
19:       $\boldsymbol{x}_{t+1}^{(1)} \leftarrow \text{Update}(\boldsymbol{x}_t^{(1)}, \eta, \hat{g}_t, s)$
20: **end for**

21: **Function** Perturb($\boldsymbol{x}, \mu, s$):
22:     Use random seed $s$ to reset random number generator
23:     **for** $x_i \in \boldsymbol{x}$ **do**
24:        $x_i \leftarrow x_i + \mu \cdot u$, where $u \sim \mathcal{N}(0, 1)$.
25:     **end for**
26:     **return** $x$

27: **Function** Update($\boldsymbol{x}, \eta, \hat{g}, s$):
28:     Use random seed $s$ to reset random number generator
29:     **for** $x_i \in \boldsymbol{x}$ **do**
30:        $x_i \leftarrow x_i - \eta \cdot \hat{g} \cdot u$, where $u \sim \mathcal{N}(0, 1)$     ▶ Standard Zeroth-Order SGD
31:     **end for**
32:     **return** $\boldsymbol{x}$

---

into two segments, $S_1$ and $S_2$, which are deployed on two GPUs $\mathcal{M}_1$ and $\mathcal{M}_2$, respectively. The detailed procedure in the $t$-th iteration is as follows:

- **Step 1:** On node $\mathcal{M}_1$, the submodel $S_1$ computes activations $a_t^+$ and $a_t$ (Lines 5-7). Here, we perturb $\boldsymbol{x}_t^{(1)}$ element-wise following (Malladi et al., 2023) to reduce memory usage. By default, we use a biased forward difference approach in Alg. 1 and distribute the unbiased central difference approach to Appendix C.2. To lower communication cost, we design a compressor $\mathbb{C}$ that first applies quantization (e.g., 4 bit) to induce high sparsity in $a_t$ and $a_t^+$ and encodes them with sparse representations (i.e., only nonzero values and their indices). Subsequently, the compressed activations $\mathbb{C}(a_t)$ and $\mathbb{C}(a_t^+)$ are transmitted to the next node $\mathcal{M}_2$.

- **Step 2:** Upon receiving $\mathbb{C}(a_t)$ and $\mathbb{C}(a_t^+)$ from $\mathcal{M}_1$, node $\mathcal{M}_2$ feed them into submodel $S_2$ to compute the ZO gradient scalar $\hat{g}_t$ (Lines 11-15). The perturbation of $\boldsymbol{x}_t^{(2)}$ is performed in the same manner as in Step 1. Then, submodel parameters at $\mathcal{M}_2$ are updated via standard ZO-SGD (Line 16). Finally, only the scalar $\hat{g}_t$ is communicated back to $\mathcal{M}_1$, instead of a high-dimensional first-order gradient, yielding substantial communication savings.

- **Step 3:** Once $\mathcal{M}_1$ receives $\hat{g}_t$, it updates sub-model parameters using the same ZO-SGD procedure applied on $\mathcal{M}_2$ (Line 19).

Here, we highlight the comparison of communication costs between central and forward difference methods. Given $P$ perturbations, in each round, utilizing the central difference method transmits $2P$ activations and $P$ gradient scalars, whereas utilizing the forward difference method transmits only $P + 1$ activations and $P$ gradient scalars. Table 2 empirically shows that the forward difference method achieves lower communication costs while maintaining comparable test accuracy compared to the central difference method.

In Fig. 3, we illustrate the execution timelines of our ZO method and conventional FO methods. The critical difference lies in the backward communication pattern: FO methods require layer-wise transmission of high-dimensional gradients during backpropagation, incurring substantial communication and time costs. In contrast, our ZO method eliminates gradient tensors entirely and instead communicates only a single scalar per iteration, which can be broadcast to all nodes simultaneously. This design not only reduces the per-iteration communication complexity from dimension-dependent to dimension-free, but also eliminates sequential gradient exchanges, thereby reducing communication cost and shortening the overall execution timeline.

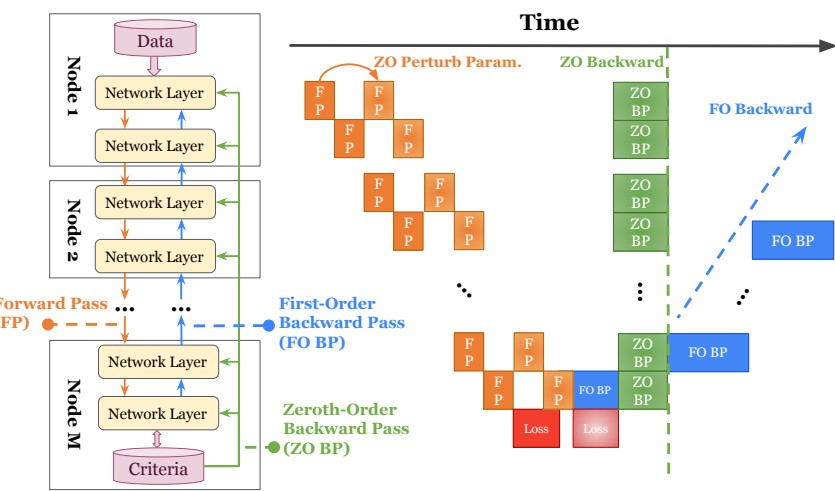

Figure 3: Execution Timeline Comparison of Zeroth-Order (e.g., `SparQ`) and First-Order Methods.

## 4 THEORETICAL ANALYSIS

We start with all assumptions used in this work. Note that $f(\cdot)$ is the loss function, $\nabla f(\cdot)$ is the first-order gradient, $\hat{\nabla} f(\cdot)$ is the zeroth-order gradient. For simplicity, we only show two-node case in the following main paper and distribute the $M > 2$ case to Appendix D.3.

**Assumption 1 (Unbiased Function Estimation)** $\mathbb{E}\left[f(\xi_t; \boldsymbol{x})\right] = f(\boldsymbol{x})$.

**Assumption 2 (Unbiased Zeroth-Order Stochastic Gradients with Bounded Variance)** *The stochastic gradient $\hat{\nabla} f(\boldsymbol{x}_k; \xi_k)$ is unbiased, and its variance is bounded, so we have*

$$\mathbb{E}[\hat{\nabla} f(\xi_k; \boldsymbol{x}_k)] = \nabla f(\boldsymbol{x}_k) \quad and \quad \mathbb{E}[\|\hat{\nabla} f(\xi_k; \boldsymbol{x}_k) - \nabla f(\boldsymbol{x}_k)\|^2] \leq \sigma^2. \tag{6}$$

**Assumption 3 (Gradient Lipschitz Condition)** *For the loss function and each composition function, their gradients are Lipschitz-continuous such that for any $\xi_t$,*

$$\|\nabla f(\xi_t; \boldsymbol{x}) - \nabla f(\xi_t; \boldsymbol{y})\| \leq L\|\boldsymbol{x} - \boldsymbol{y}\|$$

$$\left\|\nabla\left(S_1|_{\boldsymbol{x}^{(1)}}\right)(\xi_t) - \nabla\left(S_1|_{\boldsymbol{y}^{(1)}}\right)(\xi_t)\right\| \leq L_1\|\boldsymbol{x}^{(1)} - \boldsymbol{y}^{(1)}\|$$

$$\left\|\nabla\left(F \circ S_2|_{\boldsymbol{x}^{(2)}}\right)(\zeta_t) - \nabla\left(F \circ S_2|_{\boldsymbol{y}^{(2)}}\right)(\zeta_t)\right\| \leq L_2\|\boldsymbol{x}^{(2)} - \boldsymbol{y}^{(2)}\|$$

*Further, the gradients are bounded: $\left\|\nabla\left(S_1|_{\boldsymbol{x}^{(1)}}\right)(\xi_t)\right\| \leq C_{S_1}$ and $\left\|\nabla\left(F \circ S_2|_{\boldsymbol{x}^{(2)}}\right)(\zeta_t)\right\| \leq C_{S_2}$.*

**Assumption 4 (Bounded Output of Split Layer)** $\|S_1(\xi; \boldsymbol{x})\| \leq L_{S_1}, \forall \boldsymbol{x}, \xi$.

**Assumption 5 (Unbiased Compressor $\mathbb{C}(\cdot)$)** $\|\boldsymbol{x} - \mathbb{C}(\boldsymbol{x})\| \leq \kappa\|\boldsymbol{x}\|$, *where $\kappa \in [0, 1]$.*

Under the assumptions above, we derive the lemma and obtain the `SparQ`'s nonconvex convergence bound the two-node case as follows.

**Lemma 1 (Distance between Gradients of Uncompressed and Compressed Activations)** *For Algorithm 1 with $M = 2$, the difference between gradients of uncompressed and compressed activations can be bounded as: $\|\nabla f(\boldsymbol{x}_t) - \nabla \hat{f}(\boldsymbol{x}_t)\|^2 \leq (1 + C_{S_1}^2)L_2^2\kappa^2 L_{S_1}^2$.*

**Remark 1** *From Lemma 1, we observe that the upper bound is a constant, independent of learning rate $\eta$ and smoothing parameter $\mu$. This constant bound could, in principle, be further reduced. For example, AQ-SGD (Wang et al., 2022) applies a error-feedback technique on the compressed activation, but its memory cost is huge since it requires storing the previous information for each data sample on both nodes, which is mostly infeasible in practice. Our paper considers the memory limitation scenario, so we opt to compress the activation directly without any extra memory cost.*

**Theorem 1 (Convergence of `SparQ` under Non-Convexity, $M = 2$)** *Under the assumptions 1, 2, 3, 4 and 5, supposing that $\eta = \mathcal{O}(1/\sqrt{Td})$, $\mu \leq 1/(d+6)\sqrt{T}$ and $D = f(\boldsymbol{x}_0) - f(\boldsymbol{x}^\star)$, then the sequence $\{\boldsymbol{x}_t\}$ generated by `SparQ` satisfies*

$$\frac{1}{T}\sum_{t=0}^{T-1}\mathbb{E}\|\nabla f(\boldsymbol{x}_t)\|^2 = \mathcal{O}\left(D\sqrt{d/T}\right) + \mathcal{O}\left(\sqrt{d/T}\sigma^2\right) + \mathcal{O}\left((1 + C_{S_1}^2)L_2^2\kappa^2 L_{S_1}^2\sqrt{d/T}\right).$$

**Remark 2** *The first term in the above bound is associated with the distance between the initial point and the optimal point. The second term is related to the variance of stochastic gradients. Both terms have the $O(\sqrt{d/T})$ convergence rate matching with the standard ZO method in the non-convex scenario. The third one is an extra term caused by the extra compression in the activation. It does not impact the overall convergence rate asymptotically. When using a lossless compressor ($\kappa = 0$) like sparse representation, then the third term disappears, and `SparQ` can achieve a convergence rate of $\mathcal{O}(\sqrt{d/T})$ under the non-convex condition.*

## 5 EXPERIMENTS

**Models & Datasets**. In our evaluation, we utilize three NLP datasets: SST2 (Socher et al., 2013) for sentiment classification, WIC (Pilehvar & Camacho-Collados, 2019) for context-sensitive word embeddings evaluation and RTE (Bowman et al., 2015) for textual entailment recognition. For each of them, we fine-tune OPT-125M, OPT-1.3B, OPT-6.7B, GPT2 and Llama-1B models and monitor their test accuracy, peak GPU memory usage and communication cost.

**Baselines.** To comprehensively evaluate the performance, we compare `SparQ` with several baselines: first-order SGD (FO-SGD), AQ-SGD (Wang et al., 2022) and ZO-SGD (i.e., MeZO (Malladi et al., 2023)). FO-SGD represents a centralized first-order method. AQ-SGD represents a first-order model parallel method with activation (using error feedback) and gradient compression. MeZO represents a recent memory-efficient ZO method without considering communication cost. For compression level, we use 4-bit quantization to compress activations for AQ-SGD and `SparQ`, and we employ 8-bit quantization to compress backward gradients for AQ-SGD.

**Ablation Experiments.** 1) splitting between blocks and splitting immediately after activation functions to explore the impact of split positions in Fig. 4; 2) using different quantization degrees (e.g., 1-bit, 2-bit, 4-bit and 8-bit) to investigate the influence of quantization levels in Table. 1.

EXPERIMENT RESULTS:

**Split Between Blocks v.s. Split Immediately After Activation Functions.** Fig. 4 shows our splitting strategy. As shown in (a)-(c), placing the split after activation functions leads to trainable models. Specifically, ReLU induces natural sparsity that is well-preserved after 4-bit quantization, while SwiGLU and GELU, though producing dense activations, still maintain sufficient information for effective training. In contrast, (d) shows that splitting between blocks severely damages the activation representation after quantization, making the model untrainable. These results demonstrate that our splitting strategy (a-c) effectively balances quantization and trainability, validating the design choice.

**`SparQ`'s Performance with Different Quantization Levels.** Table 1 reports `SparQ`'s performance under four quantization schemes: 1-bit, 2-bit, 4-bit and 8-bit, under a single split setting ($M = 2$). We make two key observations: (1) stronger quantization (fewer bits) consistently reduces communication overhead; (2) accuracy degradation is relatively minor, even under aggressive quantization. It is worth

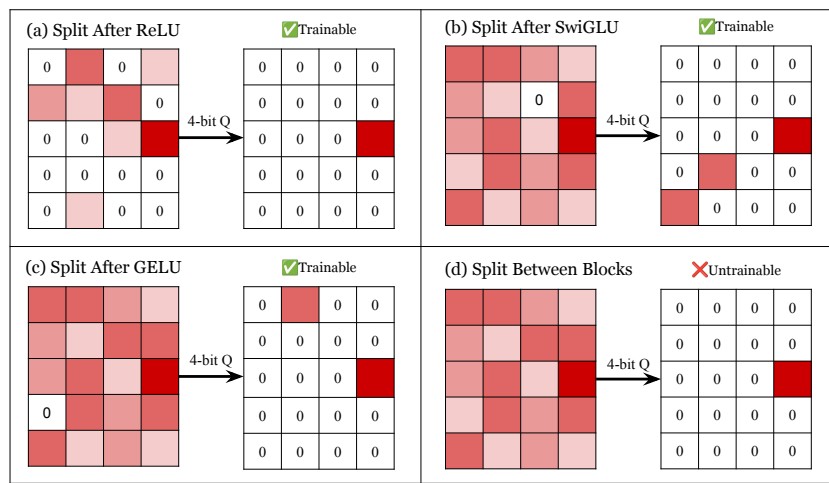

Figure 4: An Illustration of 4-Bit Quantized Activations Comparison. For each strategy, the matrix on the left represents the forward pass (e.g., activation).

noting that these results reflect the impact of quantization in the case of a single model split. When the model is split into multiple segments, the effect of quantization on both accuracy and communication overhead becomes more pronounced. Balancing accuracy and efficiency, we therefore select 4-bit quantization as the default configuration in subsequent experiments.

Table 1: Activation Quantization Levels' Impact on `SparQ`'s Test Accuracy and Communication Overhead. Hyper-parameter setup: batch size=32, $M = 2$.

| Model | Dataset | 1-bit Quantization | 2-bit Quantization | 4-bit Quantization | 8-bit Quantization |
|---|---|---|---|---|---|
| OPT-1.3B (ReLU) | SST-2 | 89.84% (0.31 GB) | 89.97% (0.58 GB) | 92.34% (1.20 GB) | 92.04% (2.32 GB) |
| | WIC | 54.75% (0.78 GB) | 55.50% (1.57 GB) | 55.62% (3.23 GB) | 56.07% (6.26 GB) |
| | RTE | 57.03% (2.49 GB) | 57.54% (4.34 GB) | 57.13% (7.18 GB) | 58.25% (14.06 GB) |
| GPT2 (GELU) | SST-2 | 84.04% (0.52 GB) | 84.15% (0.91 GB) | 84.82% (1.55 GB) | 85.12% (2.64 GB) |
| | WIC | 51.59% (1.18 GB) | 52.20% (2.33 GB) | 52.70% (3.79 GB) | 53.05% (6.88 GB) |
| | RTE | 51.26% (3.11 GB) | 52.01% (5.48 GB) | 52.37% (8.93 GB) | 52.78% (14.72 GB) |
| Llama-1B (SwiGLU) | SST-2 | 89.32% (0.65 GB) | 92.13% (1.47 GB) | 93.44% (2.42 GB) | 93.69% (5.64 GB) |
| | WIC | 52.77% (1.89 GB) | 53.35% (3.53 GB) | 53.63% (7.31 GB) | 53.82% (10.22 GB) |
| | RTE | 53.62% (3.47 GB) | 54.58% (6.83 GB) | 55.21% (12.46 GB) | 56.17% (24.74 GB) |

**Test Accuracy of `SparQ` with 4-Bit Quantization Matches ZO-SGD.** Table 2 indicates that FO-SGD achieves the highest test accuracy among all methods. Notably, `SparQ` with 4-bit quantization attains performance comparable to ZO-SGD and even outperforms AQ-SGD on the SST2 and WIC.

**`SparQ` Achieves the Best Efficiency in Both Memory and Communication.** As reported in Table 2, `SparQ` achieves the lowest overhead when considering both memory and communication costs. On the memory side, compared with FO-SGD, `SparQ` reduces total peak usage by about $30 \sim 70\%$, while also avoiding the substantial extra per-sample storage required by AQ-SGD. This shows that `SparQ` can effectively scale to larger models where FO methods often encounter memory bottlenecks. On the communication side, `SparQ` stably outperforms both FO-SGD and AQ-SGD, with the forward-difference variant consistently achieving the lowest communication cost across all model scales. On average, `SparQ` reduces communication overhead by more than 50% compared to AQ-SGD and over 70% compared to FO-SGD. Taken together, these results establish `SparQ` as the most efficient method overall, striking a favorable balance between maintaining accuracy and minimizing both memory and communication demands.

# 6 RELATED WORK

**Model Parallelism (MP).** MP (Dean et al., 2012) is a fundamental technique in distributed deep learning that partitions a deep neural network into disjoint segments, each assigned to a separate computing node (e.g., a GPU or machine). Building on MP, various algorithms have been proposed. Among these, AQ-SGD (Wang et al., 2022) is particularly relevant to this work. It employs pipeline parallelism and can function effectively over slow networks but introduces a huge extra memory cost.

Table 2: Test Accuracy, Memory Cost, and Communication Cost. 1) Memory cost here means total peak memory usage. For AQ-SGD, the value before $+$ is the total peak message usage on two GPUs, and the value after $+$ is the extra memory usage to store per-sample messages, which are on SSD or CPU, as Wang et al. (2022); 2) Key hyper-parameter setup: $P = 5$ for all ZO methods, batch size$= 32$; 3) "4Q" means 4-bit quantization. "forward" means forward difference method.

| Model | Dataset | FO-SGD | AQ-SGD | ZO-SGD | Ours (4Q, forward) |
|-------|---------|--------|--------|--------|--------------------|
| OPT-125M | SST-2 | 87.5% (2 GB, 2.8 GB) | 82.3% (2+11 GB, 1.1 GB) | 85.2% (1 GB, 3.1 GB) | 84.5% (1 GB, **0.5 GB**) |
| | WIC | 54.1% (3 GB, 7.2 GB) | 53.9% (3+5 GB, 2.7 GB) | 53.6% (2 GB, 8.3 GB) | 53.4% (2 GB, **1.2 GB**) |
| | RTE | 56.5% (11 GB, 15.7 GB) | 54.3% (11+10 GB, 5.9 GB) | 53.5% (6 GB, 17.1 GB) | 53.2% (6 GB, **2.6 GB**) |
| OPT-1.3B | SST-2 | 91.7% (15 GB, 8 GB) | 84.2% (15+31 GB, 3 GB) | 90.6% (6 GB, 8 GB) | 92.3% (6 GB, **1 GB**) |
| | WIC | 63.5% (16 GB, 19 GB) | 56.7% (16+9 GB, 7 GB) | 55.8% (7 GB, 21 GB) | 55.6% (7 GB, **3 GB**) |
| | RTE | 70.8% (41 GB, 42 GB) | 60.2% (41+34 GB, 16 GB) | 57.3% (11 GB, 47 GB) | 57.1% (11 GB, **7 GB**) |
| OPT-6.7B | SST-2 | 94.4% (76 GB, 15 GB) | 89.6% (76+61 GB, 6 GB) | 92.1% (26 GB, 11 GB) | 92.0% (26 GB, **2 GB**) |
| | WIC | 65.8% (78 GB, 39 GB) | 56.8% (78+19 GB, 15 GB) | 58.7% (27 GB, 31 GB) | 57.8% (27 GB, **5 GB**) |
| | RTE | 71.1% (202 GB, 83 GB) | 64.3% (202+69 GB, 31 GB) | 63.5% (29 GB, 63 GB) | 63.1% (29 GB, **9 GB**) |
| GPT2 | SST-2 | 88.1% (2 GB, 4 GB) | 84.5% (2+10 GB, 3 GB) | 84.9% (1 GB, 4 GB) | 84.8% (1 GB, **2 GB**) |
| | WIC | 61.3% (2 GB, 22 GB) | 55.6% (2+5 GB, 9 GB) | 52.5% (1 GB, 25 GB) | 52.7% (1 GB, **4 GB**) |
| | RTE | 63.1% (5 GB, 46 GB) | 55.9% (5+10 GB, 20 GB) | 52.3% (3 GB, 52 GB) | 52.3% (3 GB, **9 GB**) |
| Llama-1B | SST-2 | 94.3% (14 GB, 8 GB) | 93.2% (14+30 GB, 4 GB) | 93.7% (5 GB, 11 GB) | 93.4% (5 GB, **2 GB**) |
| | WIC | 60.4% (16 GB, 18 GB) | 55.7% (16+10 GB, 8 GB) | 53.6% (6 GB, 20 GB) | 53.6% (6 GB, **7 GB**) |
| | RTE | 64.7% (25 GB, 41 GB) | 59.1% (25+30 GB, 17 GB) | 55.8% (8 GB, 42 GB) | 55.2% (8 GB, **12 GB**) |

**Zeroth-Order (ZO) Optimization.** ZO optimization is a gradient-free approach that estimates gradients using only differences in function values and random perturbation vectors (Liu et al., 2020), in contrast to first-order methods, which rely on explicitly computed gradients. Prior research has demonstrated the efficacy of ZO optimization in black-box attack (Kariyappa et al., 2021; Yu et al., 2024), reinforcement learning (Pan et al., 2022; Jing et al., 2024), communication savings (Fang et al., 2022; Qin et al., 2024; Li et al., 2025a;b), etc. In addition, ZO optimization has been shown to lower memory consumption during LLM fine-tuning. For example, MeZO (Malladi et al., 2023) and LOZO (Chen et al., 2025) utilize ZO optimization to perform solely forward passes, thereby eliminating the need to store gradients from backward propagation. Yet, the integration of ZO optimization within MP frameworks remains largely unexplored.

**Activation Compression & Sparsity.** Compression is widely adopted to reduce communication and memory overhead in distributed systems. Popular compression techniques, such as quantization (Gray & Neuhoff, 1998; Alistarh et al., 2017; Horváth et al., 2023; Lin et al., 2024) and sparsification (Wangni et al., 2018; Yang et al., 2021; Yoon & Oh, 2023), have primarily targeted gradients and weights. Recently, attention has shifted toward compressing activations (Evans & Aamodt, 2021; Liu et al., 2021; 2022; Chen et al., 2021; Eliassen & Selvan, 2024; Lin et al., 2024). For example, ActNN (Chen et al., 2021) and ALAM (Woo et al., 2024) quantize activations to improve memory efficiency. Moreover, AQ-SGD (Wang et al., 2022) quantizes backward gradients and the changes of forward activation to enable communication-efficient training in pipeline parallel architectures. However, its reliance on error-feedback mechanism to guarantee convergence necessitates storing per-sample information on CPUs or SSDs, thereby imposing a significant extra memory burden.

## 7 CONCLUSION

In summary, we introduce `SparQ`, a ZO model-parallel framework with split layer allocation informed by quantization-induced activation sparsity, reducing communication overhead and minimizes memory costs for LLM fine-tuning under MP. Our theoretical analysis establishes a sublinear convergence rate in non-convex settings, and empirical results show that `SparQ` reduces GPU memory consumption by more than $3\times$ and communication cost by over $50\%$ compared to state-of-the-art baselines. These findings highlight the potential of ZO, sparsity-guided frameworks for scaling LLM fine-tuning, and open up promising directions for future research on distributed optimization.

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

APPENDIX

CONTENT

## A    STATEMENT OF LLM USAGE

During the preparation of this manuscript, we used LLM tools to help with language refinement and stylistic improvements. After each use of the tool, we carefully reviewed and validated the correctness and appropriateness of the generated text to ensure accuracy and alignment with the intended meaning.

## B    ADDITIONAL EXPERIMENT DETAILS AND RESULTS

### B.1    SPLIT LAYER SELECTION

In the two-node ($M = 2$) setting, to balance the computation and memory burden across the two nodes, we select the split layer near the middle block where the quantized output exhibits the highest sparsity. Accordingly, in our experiments, we split the OPT models immediately after the activation function in the middle block (e.g., Block 6 for OPT-125M), the Llama-1B model after the activation function in Block 15, and the GPT-2 model after the activation function in Block 7.

### B.2    GPU PEAK MEMORY USAGE

We execute a group of experiments to test GPU peak memory usages by fine-tuning the SST2 dataset across OPT-1.3B, OPT-2.7B, OPT-6.7B and OPT-13B models. Except for different model sizes, all hyperparameter setups are the same. Fig. 5 reveals several key empirical observations:

1) The memory overhead for first-order methods is over $3\times$ more than that for memory-efficient zeroth-order methods (e.g., MeZO and `SparQ`).

2) The memory cost for `SparQ` is approximately equal to the model size, matching the conclusion in (Malladi et al., 2023) and showing the significant memory reduction compared with first-order methods.

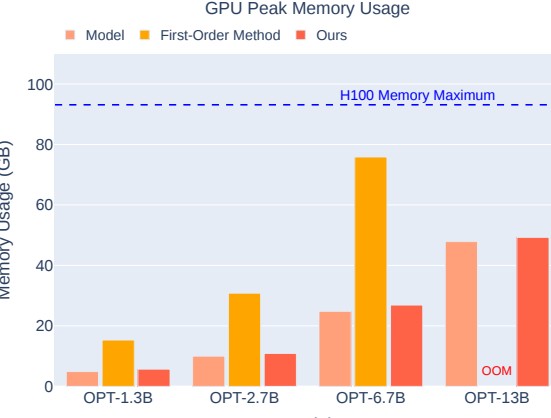

Figure 5: GPU Peak Memory Usage across Multiple LLMs When the Fine-tuning SST-2 Dataset. "OOM" means out of memory. Experiment setup: train batch size=test batch size= 32, momentum= 0.

## C    SPARQ'S TECHNICAL DETAILS AND FRAMEWORK EXTENSIONS

### C.1    WHY ACTIVATION SPARSITY CAN BE HELPFUL TO REDUCE COMMUNICATION COST?

The key intuition behind using sparse representations is that activations after quantization often contain a large fraction of zeros. Instead of transmitting the full dense tensors, we can **encode only the non-zero values together with their corresponding indices**. This compressed form eliminates the need to communicate redundant zero entries, thereby drastically reducing the volume

of data transmitted across devices. From a theoretical perspective, if the activation sparsity ratio is $\rho$ (i.e., only $\rho$ fraction of entries remain non-zero), then the effective communication cost scales proportionally to $\rho d$ rather than the full dimensionality $d$. In practice, modern LLM activations exhibit high sparsity under low-bit quantization, which makes sparse representations particularly effective. By combining quantization-induced sparsity with index-based encoding, `SparQ` achieves highly compressed communication while preserving task accuracy.

### C.2   SParQ with Central Difference Method

In our main paper, we focus on the biased forward difference method to estimate zeroth-order gradients because of its advantage of less activation transmission and comparable precision performance. Here, we introduce another widely used gradient estimation approach - the central difference method, which estimates the ZO gradient as:

$$\hat{G} = \frac{1}{P} \sum_{i=1}^{P} g_i \cdot \boldsymbol{u}_i = \frac{1}{P} \sum_{i=1}^{P} \frac{f(\xi; \boldsymbol{x} + \mu \boldsymbol{u}_i) - f(\xi; \boldsymbol{x} - \mu \boldsymbol{u}_i)}{2\mu} \cdot \boldsymbol{u}_i, \tag{7}$$

Additionally, when integrating ZOO using the unbiased central difference method into model parallelism formulation, the expression in Eq. (4) of computing the zeroth-order gradient scalar will be replaced by

$$g_{i,t} = \frac{F(a_{i,t}^+; \boldsymbol{x}_t^{(2)} + \mu \boldsymbol{u}_{i,t}^{(2)}) - F(a_{i,t}^-; \boldsymbol{x}_t^{(2)} - \mu \boldsymbol{u}_{i,t}^{(2)})}{2\mu}, \tag{8}$$

where activations are computed by

$$a_{i,t}^+ = S_1\big(\xi_t \,;\, \boldsymbol{x}_t^{(1)} + \mu \boldsymbol{u}_{i,t}^{(1)}\big), \ \ a_{i,t}^- = S_1\big(\xi_t \,;\, \boldsymbol{x}_t^{(1)} - \mu \boldsymbol{u}_{i,t}^{(1)}\big). \tag{9}$$

In our experiments, the main performance difference of forward and central difference methods lies in communication overhead. Overall, central difference incurs higher communication cost compared to forward difference, as analyzed in Sec. C.3.

### C.3   SParQ Across Multiple Computing Nodes ($M > 2$)

Our framework can also be straightforwardly extended to multiple computing node scenarios ($M > 2$). Alg. 2 demonstrates `SparQ` using the central difference method under the multi-node case ($M > 2$), and Alg. 3 shows `SparQ` using the forward difference method under the multi-node case ($M > 2$). In Sec. D.3, we provide the convergence theorem and its proof of the $M > 2$ case. The main impact of LLM fine-tuning on multiple computing nodes is the increased communication overhead attributable to the additional split layers. Nevertheless, compared with AQ-SGD (Wang et al., 2022), `SparQ` still can achieving $1 \sim 2\times$ communication savings.

**Communication Cost Analysis** ($M > 2$)**.** Given the number of computing nodes $M$, and the number of perturbations $P$, the central difference method requires transmitting $2 \times P \times (M - 1)$ activations and $P \times (M - 1)$ gradient scalars per training round, whereas the forward difference method requires transmitting only $(P + 1) \times (M - 1)$ activations and $P \times (M - 1)$ gradient scalars per round. Therefore, in general, the communication overhead of using the forward difference method is lower than the cost of using the central difference method.

## D   Proof

### D.1   Lemmas

The following Lemma 2 and Lemma 3 are relative to zeroth-order optimization and have been commonly used in zeroth-order proof. It is worth pointing out that Lemma 3 is dependent on the forward difference method, which can be known by (13). Consequently, we use it to prove the convergence of Alg. 1 ($M = 2$) and Alg. 3 ($M > 2$).

**Lemma 2** $f \in \mathcal{C}_L^{1,1}(\mathbb{R}^d)$ *if $f$ is differentiable and satisfies*

$$\|\nabla f(\boldsymbol{x}) - \nabla f(\boldsymbol{y})\| \le L \|\boldsymbol{x} - \boldsymbol{y}\|. \tag{10}$$

**Algorithm 2** SparQ ($M > 2$) with Central Difference Method

---

1: **Initialize**: split model immediately after activation functions to get submodels $S_1, ..., S_M$, model parameter $\mathbf{x}_0 = \mathrm{col}[\boldsymbol{x}_0^{(1)}, \cdots, \mathbf{x}_0^{(M)}]$, learning rate $\eta$, smoothing parameter $\mu$, the number of iterations $T$.
2: **for** $t = 0, 1, ..., T-1$ **do**
3:    On node $\mathcal{M}_1$:
4:       Sample a random seed $s$ and a data sample $\xi_t$
5:       $\boldsymbol{x}_t^{(1)} \leftarrow \text{Perturb}(\boldsymbol{x}_t^{(1)}, \mu, s)$
6:       Compute $a_t^+ = S_1(\xi_t \,; \boldsymbol{x}_t^{(1)})$
7:       $\boldsymbol{x}_t^{(1)} \leftarrow \text{Perturb}(\boldsymbol{x}_t^{(1)}, -2\mu, s)$
8:       Compute $a_t^- = S_1(\xi_t \,; \boldsymbol{x}_t^{(1)})$
9:       $\boldsymbol{x}_t^{(1)} \leftarrow \text{Perturb}(\boldsymbol{x}_t^{(1)}, \mu, s)$
10:      Send $\mathbb{C}(a_t^+)$ and $\mathbb{C}(a_t^-)$, which are sparse representations of quantized $a_t^+$ and $a_t^-$, to $\mathcal{M}_2$.

11:         Here we skip the description of all processes on $\mathcal{M}_2 \cdots \mathcal{M}_{M-1}$ because they have the similar process.
12:   On node $\mathcal{M}_M$:
13:      $\boldsymbol{x}_t^{(M)} \leftarrow \text{Perturb}(\boldsymbol{x}_t^{(M)}, \mu, s)$
14:      Compute $f_t^+ = F\big(S_M(\mathbb{C}(a_t^+)\,; \boldsymbol{x}_t^{(M)})\big)$
15:      $\boldsymbol{x}_t^{(M)} \leftarrow \text{Perturb}(\boldsymbol{x}_t^{(M)}, -2\mu, s)$
16:      Compute $f_t^- = F\big(S_M(\mathbb{C}(a_t^-)\,; \boldsymbol{x}_t^{(M)})\big)$
17:      $\boldsymbol{x}_t^{(M)} \leftarrow \text{Perturb}(\boldsymbol{x}_t^{(M)}, \mu, s)$
18:      $\hat{g}_t = \frac{1}{2\mu}(f_t^+ - f_t^-)$
19:      $\boldsymbol{x}_{t+1}^{(M)} \leftarrow \text{Update}(\boldsymbol{x}_t^{(M)}, \eta, \hat{g}_t, s)$
20:      Send $\hat{g}_t$ back to $\mathcal{M}_i$, where $i \in [1, \cdots, M-1]$.          ▶ Only a scalar is transmitted
21:   On nodes $\mathcal{M}_i$, $i \in [1, ..., M-1]$:
22:      $\boldsymbol{x}_{t+1}^{(i)} \leftarrow \text{Update}(\boldsymbol{x}_t^{(i)}, \eta, \hat{g}_t, s)$
23: **end for**

24: **Function** Perturb($\boldsymbol{x}, \mu, s$):
25:    Use random seed $s$ to reset random number generator
26:    **for** $x_i \in \boldsymbol{x}$ **do**
27:      $x_i \leftarrow x_i + \mu \cdot u$, where $u \sim \mathcal{N}(0, 1)$.
28:    **end for**
29:    **return** $x$

30: **Function** Update($\boldsymbol{x}, \eta, \hat{g}, s$):
31:    Use random seed $s$ to reset random number generator
32:    **for** $x_i \in \boldsymbol{x}$ **do**
33:      $x_i \leftarrow x_i - \eta \cdot \hat{g} \cdot u$, where $u \sim \mathcal{N}(0, 1)$.          ▶ Standard Zeroth-Order SGD
34:    **end for**
35:    **return** $\boldsymbol{x}$

---

*Also, we have*

$$|f(\boldsymbol{y}) - f(\boldsymbol{x}) - \langle \nabla f(\boldsymbol{x}), \boldsymbol{y} - \boldsymbol{x} \rangle| \leq \frac{L}{2} \|\boldsymbol{y} - \boldsymbol{x}\|^2. \tag{11}$$

**Lemma 3** *(Ghadimi & Lan, 2013; Nesterov & Spokoiny, 2017) We define a smooth approximation of objective function $f$ as $f_\mu(\cdot)$ that can be formulated as*

$$f_\mu(\boldsymbol{x}) := \frac{1}{(2\pi)^{\frac{d}{2}}} \int f(\boldsymbol{x} + \mu\boldsymbol{u}) e^{-\frac{1}{2}\|\boldsymbol{u}\|^2} d\boldsymbol{u} = \mathbb{E}\left[f(\boldsymbol{x} + \mu\boldsymbol{u})\right] \tag{12}$$

*Then, for any $f \in \mathcal{C}_L^{1,1}$, the following statements hold.*

---

**Algorithm 3** SparQ ($M > 2$) with Forward Difference Method

---

1: **Initialize**: split model immediately after activation functions to get submodels $S_1, ..., S_M$, model parameter $\mathbf{x}_0 = \text{col}[\boldsymbol{x}_0^{(1)}, \cdots, \mathbf{x}_0^{(M)}]$, learning rate $\eta$, smoothing parameter $\mu$, the number of iterations $T$.
2: **for** $t = 0, 1, ..., T - 1$ **do**
3:     On node $\mathcal{M}_1$:
4:         Sample a random seed $s$ and a data sample $\xi_t$
5:         Compute $a_t = S_1\big(\xi_t \,;\, \boldsymbol{x}_t^{(1)}\big)$
6:         $\boldsymbol{x}_t^{(1)} \leftarrow \text{Perturb}(\boldsymbol{x}_t^{(1)}, \mu, s)$
7:         Compute $a_t^+ = S_1\big(\xi_t \,;\, \boldsymbol{x}_t^{(1)}\big)$
8:         $\boldsymbol{x}_t^{(1)} \leftarrow \text{Perturb}(\boldsymbol{x}_t^{(1)}, -\mu, s)$
9:         Send $\mathbb{C}(a_t^+)$ and $\mathbb{C}(a_t)$, which are sparse representations of quantized $a_t^+$ and $a_t$, to $\mathcal{M}_2$.
10:        Here we skip the description of all processes on $\mathcal{M}_2 \cdots \mathcal{M}_{M-1}$ because they have the similar process.
11:     On node $\mathcal{M}_M$:
12:         Compute $f_t = F\big(S_M(\mathbb{C}(a_t) \,;\, \boldsymbol{x}_t^{(M)})\big)$
13:         $\boldsymbol{x}_t^{(M)} \leftarrow \text{Perturb}(\boldsymbol{x}_t^{(M)}, \mu, s)$
14:         Compute $f_t^+ = F\big(S_M(\mathbb{C}(a_t^+) \,;\, \boldsymbol{x}_t^{(M)})\big)$
15:         $\boldsymbol{x}_t^{(M)} \leftarrow \text{Perturb}(\boldsymbol{x}_t^{(M)}, -\mu, s)$
16:         $\hat{g}_t = \frac{1}{\mu}(f_t^+ - f_t)$
17:         $\boldsymbol{x}_{t+1}^{(M)} \leftarrow \text{Update}(\boldsymbol{x}_t^{(M)}, \eta, \hat{g}_t, s)$
18:         Send $\hat{g}_t$ back to $\mathcal{M}_i$, where $i \in [1, \cdots, M - 1]$.     ▶ Only a scalar is transmitted
19:     On nodes $\mathcal{M}_i, i \in [1, ..., M - 1]$:
20:         $\boldsymbol{x}_{t+1}^{(i)} \leftarrow \text{Update}(\boldsymbol{x}_t^{(i)}, \eta, \hat{g}_t, s)$
21: **end for**

22: **Function** Perturb($\boldsymbol{x}, \mu, s$):
23:     Use random seed $s$ to reset random number generator
24:     **for** $x_i \in \boldsymbol{x}$ **do**
25:         $x_i \leftarrow x_i + \mu \cdot u$, where $u \sim \mathcal{N}(0, 1)$.
26:     **end for**
27:     **return** $x$

28: **Function** Update($\boldsymbol{x}, \eta, \hat{g}, s$):
29:     Use random seed $s$ to reset random number generator
30:     **for** $x_i \in \boldsymbol{x}$ **do**
31:         $x_i \leftarrow x_i - \eta \cdot \hat{g} \cdot u$, where $u \sim \mathcal{N}(0, 1)$.     ▶ Standard Zeroth-Order SGD
32:     **end for**
33:     **return** $x$

---

*(a) The gradient of $f_\mu(\cdot)$ is $L_\mu$-Lipschitz continuous where $L_\mu \leq L$. $\nabla f_\mu(\boldsymbol{x})$ can be shown as*

$$\nabla f_\mu(\boldsymbol{x}) = \frac{1}{(2\pi)^{\frac{d}{2}}} \int \frac{f(\boldsymbol{x} + \mu \boldsymbol{u}) - f(\boldsymbol{x})}{\mu} \boldsymbol{u} e^{-\frac{1}{2}\|\boldsymbol{u}\|^2} d\boldsymbol{u}. \tag{13}$$

*(b) For any $\boldsymbol{x} \in \mathbb{R}^n$,*

$$|f_\mu(\boldsymbol{x}) - f(\boldsymbol{x})| \leq \frac{\mu^2}{2} L d \tag{14}$$

$$\|\nabla f_\mu(\boldsymbol{x}) - \nabla f(\boldsymbol{x})\| \leq \frac{\mu}{2} L (d + 3)^{\frac{3}{2}} \tag{15}$$

*(c) For any $\boldsymbol{x} \in \mathbb{R}^n$,*

$$\frac{1}{\mu^2} \mathbb{E}_{\boldsymbol{u}} \left[ \big(f(\boldsymbol{x} + \mu \boldsymbol{u}) - f(\boldsymbol{x})\big)^2 \|\boldsymbol{u}\|^2 \right] \leq \frac{\mu^2}{2} L^2 (d + 6)^3 + 2(d + 4)\|\nabla f(\boldsymbol{x})\|^2 \tag{16}$$

Following form (15) and utilizing Jensen's inequality $\|a\|^2 \leq 2\|a - b\|^2 + 2\|b\|^2$, we have

$$\|\nabla f_\mu(\boldsymbol{x})\|^2 \leq 2\|\nabla f(\boldsymbol{x})\|^2 + \frac{\mu^2}{2}L^2(d+3)^3, \tag{17}$$

$$\|\nabla f(\boldsymbol{x})\|^2 \leq 2\|\nabla f_\mu(\boldsymbol{x})\|^2 + \frac{\mu^2}{2}L^2(d+3)^3. \tag{18}$$

Moreover, we denote $f_\mu^* := \min_{\boldsymbol{x} \in \mathbb{R}^d} f_\mu(\boldsymbol{x})$ and conclude $|f_\mu^* - f^*| \leq \frac{\mu^2 Ld}{2}$ from (14). Then, we further conclude that

$$-\mu^2 Ld \leq \left[f_\mu(\boldsymbol{x}) - f_\mu^*\right] - \left[f(\boldsymbol{x}) - f^*\right] \leq \mu^2 Ld \tag{19}$$

**Lemma 4 (Distance between Gradients of Uncompressed and Compressed Activations, $M = 2$)**
*For Alg. 1, the difference between gradients of uncompressed and compressed activations can be bounded by a constant term as follows:*

$$\left\|\nabla f(\boldsymbol{x}_t) - \nabla \hat{f}(\boldsymbol{x}_t)\right\|^2 \leq (1 + C_{S_1}^2)L_2^2\kappa^2 L_{S_1}^2.$$

*Proof of Lemma 4:*

$$\left\|\nabla f(\boldsymbol{x}_t) - \nabla \hat{f}(\boldsymbol{x}_t)\right\|^2 = \|\nabla_{\boldsymbol{x}^{(1)}} f(\boldsymbol{x}_t) - \nabla_{\boldsymbol{x}^{(1)}} \hat{f}(\boldsymbol{x}_t)\|^2 + \|\nabla_{\boldsymbol{x}^{(2)}} f(\boldsymbol{x}_t) - \nabla_{\boldsymbol{x}^{(2)}} \hat{f}(\boldsymbol{x}_t)\|^2$$

$$= \left\|\nabla_{\boldsymbol{x}^{(1)}}(F \circ S_2 \circ S_1)\big|_{(\boldsymbol{x}_t^{(1)}, \boldsymbol{x}_t^{(2)})} - \nabla_{S_1}(F \circ S_2)\big|_{(\mathbb{C}(S_1(\xi; \boldsymbol{x}_t^{(1)})), \boldsymbol{x}_t^{(2)})} \cdot \nabla_{\boldsymbol{x}^{(1)}} S_1\big|_{\boldsymbol{x}_t^{(1)}}\right\|^2$$

$$+ \left\|\nabla_{\boldsymbol{x}^{(2)}}(F \circ S_2)\big|_{(S_1(\xi; \boldsymbol{x}_t^{(1)}), \boldsymbol{x}_t^{(2)})} - \nabla_{\boldsymbol{x}^{(2)}}(F \circ S_2)\big|_{(\mathbb{C}(S_1(\xi; \boldsymbol{x}_t^{(1)})), \boldsymbol{x}_t^{(2)})}\right\|^2$$

$$\leq C_{S_1}^2 L_2^2 \left\|(\mathbb{C}(S_1(\xi, \boldsymbol{x}_t^{(1)})), \boldsymbol{x}_t^{(2)}) - (S_1(\xi, \boldsymbol{x}_t^{(1)}), \boldsymbol{x}_t^{(2)})\right\|^2$$

$$+ L_2^2 \left\|(\mathbb{C}(S_1(\xi; \boldsymbol{x}_t^{(1)})); \boldsymbol{x}_t^{(2)}) - (S_1(\xi; \boldsymbol{x}_t^{(1)}); \boldsymbol{x}_t^{(2)})\right\|^2$$

$$= (1 + C_{S_1}^2)L_2^2 \left\|(\mathbb{C}(S_1(\xi; \boldsymbol{x}_t^{(1)})); \boldsymbol{x}_t^{(2)}) - (S_1(\xi; \boldsymbol{x}_t^{(1)}); \boldsymbol{x}_t^{(2)})\right\|^2$$

$$\leq (1 + C_{S_1}^2)L_2^2\kappa^2 \left\|S_1(\xi; \boldsymbol{x}_t^{(1)})\right\|^2$$

$$\leq (1 + C_{S_1}^2)L_2^2\kappa^2 L_{S_1}^2, \tag{20}$$

where we get (20) by assumption 4. ∎

## D.2 PROOF OF THEOREM 1

Before showing the proof of the theorem, we present notations, definitions and update rules used in the following proof.

$$\boldsymbol{x}_{t+1}^{(1)} = \boldsymbol{x}_t^{(1)} - \eta \cdot \hat{G}_\mu(\boldsymbol{x}_t, \xi_t, \boldsymbol{u}_t) = \boldsymbol{x}_t^{(1)} - \eta \cdot \hat{g}_\mu(\boldsymbol{x}_t, \xi_t, \boldsymbol{u}_t) \cdot \boldsymbol{u}_t^{(1)} \tag{21}$$

$$\boldsymbol{x}_{t+1}^{(2)} = \boldsymbol{x}_t^{(2)} - \eta \cdot \hat{G}_\mu(\boldsymbol{x}_t, \xi_t, \boldsymbol{u}_t) = \boldsymbol{x}_t^{(2)} - \eta \cdot \hat{g}_\mu(\boldsymbol{x}_t, \xi_t, \boldsymbol{u}_t) \cdot \boldsymbol{u}_t^{(2)} \tag{22}$$

Introducing $\boldsymbol{x}_t = \text{col}[\boldsymbol{x}_t^{(1)}, \boldsymbol{x}_t^{(2)}]$ and $\boldsymbol{u}_t = \text{col}[\boldsymbol{u}_t^{(1)}, \boldsymbol{u}_t^{(2)}]$, we can write it into one line

$$\boldsymbol{x}_{t+1} = \boldsymbol{x}_t - \eta \cdot g_\mu(\boldsymbol{x}_t, \xi_t, \boldsymbol{u}_t) \cdot \boldsymbol{u}_t \tag{23}$$

Next, we define a new (virtual) zeroth-order gradient scalar without the compression step:

$$g_\mu(\boldsymbol{x}_t, \xi_t, \boldsymbol{u}_t)$$

$$= \frac{F\big(S_2(a_t^+; \boldsymbol{x}_t^{(2)} + \mu \boldsymbol{u}_t^{(2)})\big) - F\big(S_2(a_t^-; \boldsymbol{x}_t^{(2)} - \mu \boldsymbol{u}_t^{(2)})\big)}{\mu}$$

$$= \frac{F\big(S_2(S_1(\xi_t; \boldsymbol{x}_t^{(1)} + \mu \boldsymbol{u}_t^{(1)}); \boldsymbol{x}_t^{(2)} + \mu \boldsymbol{u}_t^{(2)})\big) - F\big(S_2(S_1(\xi_t; \boldsymbol{x}_t^{(1)} - \mu \boldsymbol{u}_t^{(1)}); \boldsymbol{x}_t^{(2)} - \mu \boldsymbol{u}_t^{(2)})\big)}{\mu}$$

$$\tag{24}$$

It is not hard to see that $\mathbb{E}_{\xi_t, \boldsymbol{u}_t}[G_\mu(\boldsymbol{x}_t, \xi_t, \boldsymbol{u}_t)] = \nabla f_\mu(\boldsymbol{x}_t)$ using the Lemma 3 (a). Moreover, we define the difference between these two gradient scalars as:

$$\Delta_\mu(\boldsymbol{x}_t, \xi_t, \boldsymbol{u}_t) := g_\mu(\boldsymbol{x}_t, \xi_t, \boldsymbol{u}_t) - \hat{g}_\mu(\boldsymbol{x}_t, \xi_t, \boldsymbol{u}_t) \tag{25}$$

Now, we arrive at

$$\boldsymbol{x}_{t+1} = \boldsymbol{x}_t - \eta g_\mu(\boldsymbol{x}_t, \xi_t, \boldsymbol{u}_t) \cdot \boldsymbol{u}_t + \eta \Delta_\mu(\boldsymbol{x}_t, \xi_t, \boldsymbol{u}_t) \cdot \boldsymbol{u}_t \tag{26}$$

$$= \boldsymbol{x}_t - \eta G_\mu(\boldsymbol{x}_t, \xi_t, \boldsymbol{u}_t) + \eta \Delta_\mu(\boldsymbol{x}_t, \xi_t, \boldsymbol{u}_t) \cdot \boldsymbol{u}_t \tag{27}$$

When there is no ambiguities, we simply use $G_\mu$ and $\Delta_\mu$ to replace $G_\mu(\boldsymbol{x}_t, \xi_t, \boldsymbol{u}_t)$ and $\Delta_\mu(\boldsymbol{x}_t, \xi_t, \boldsymbol{u}_t)$ respectively.

*Proof of Theorem 1:*

We start with the Lipschitz condition:

$$\mathbb{E}\left[f_\mu(\boldsymbol{x}_{t+1})\right]$$

$$\leq f_\mu(\boldsymbol{x}_t) + \mathbb{E}\left[\langle \nabla f_\mu(\boldsymbol{x}_t), \boldsymbol{x}_{t+1} - \boldsymbol{x}_t \rangle\right] + \frac{L}{2}\mathbb{E}\|\boldsymbol{x}_{t+1} - \boldsymbol{x}_t\|^2$$

$$= f_\mu(\boldsymbol{x}_t) - \eta\langle \nabla f_\mu(\boldsymbol{x}_t), \mathbb{E}\left[G_\mu - \Delta_\mu \cdot \boldsymbol{u}_t\right]\rangle + \frac{L}{2}\eta^2 \mathbb{E}\left\|\hat{G}_\mu\right\|^2$$

$$\leq f_\mu(\boldsymbol{x}_t) - \eta\mathbb{E}\left[\langle \nabla f_\mu(\boldsymbol{x}_t), G_\mu \rangle\right] + \frac{\eta}{2\epsilon}\mathbb{E}\|\nabla f_\mu(\boldsymbol{x}_t)\|^2 + \frac{\eta\epsilon}{2}\left\|\mathbb{E}\left[\Delta_\mu \cdot \boldsymbol{u}_t\right]\right\|^2 + L\eta^2\mathbb{E}\left\|\hat{G}_\mu\right\|^2 \tag{28}$$

$$= f_\mu(\boldsymbol{x}_t) - \eta\mathbb{E}\left[\langle \nabla f_\mu(\boldsymbol{x}_t), \nabla f_\mu(\boldsymbol{x}_t) - \nabla f_\mu(\boldsymbol{x}_t) + G_\mu \rangle\right] + \frac{\eta}{2\epsilon}\mathbb{E}\|\nabla f_\mu(\boldsymbol{x}_t)\|^2$$

$$+ \frac{\eta\epsilon}{2}\left\|\mathbb{E}\left[G_\mu - \hat{G}_\mu\right]\right\|^2 + L\eta^2\mathbb{E}\left\|\hat{G}_\mu\right\|^2$$

$$= f_\mu(\boldsymbol{x}_t) - \eta\mathbb{E}\|\nabla f_\mu(\boldsymbol{x}_t)\|^2 - \eta\mathbb{E}\left[\langle \nabla f_\mu(\boldsymbol{x}_t), G_\mu - \nabla f_\mu(\boldsymbol{x}_t) \rangle\right] + \frac{\eta}{2\epsilon}\mathbb{E}\|\nabla f_\mu(\boldsymbol{x}_t)\|^2$$

$$+ \frac{\eta\epsilon}{2}\left\|\nabla f_\mu(\boldsymbol{x}_t) - \nabla\hat{f}_\mu(\boldsymbol{x}_t)\right\|^2 + L\eta^2\mathbb{E}\left\|\hat{G}_\mu\right\|^2$$

$$= f_\mu(\boldsymbol{x}_t) + (\frac{\eta}{2\epsilon} - \eta)\mathbb{E}\|\nabla f_\mu(\boldsymbol{x}_t)\|^2 + \frac{\eta\epsilon}{2}\left\|\nabla f_\mu(\boldsymbol{x}_t) - \nabla\hat{f}_\mu(\boldsymbol{x}_t)\right\|^2 + L\eta^2\mathbb{E}\left\|\hat{G}_\mu\right\|^2, \tag{29}$$

where (28) applies the Young's inequality on the cross term, where $\epsilon$ is an arbitrary positive number, and the Jensen's inequality on the last term.

Re-arranging (29), we get

$$\eta(1 - \frac{1}{2\epsilon})\mathbb{E}\|\nabla f_\mu(\boldsymbol{x}_t)\|^2 \leq f_\mu(\boldsymbol{x}_t) - \mathbb{E}\left[f_\mu(\boldsymbol{x}_{t+1})\right] + \frac{\eta\epsilon}{2}\left\|\nabla f_\mu(\boldsymbol{x}_t) - \nabla\hat{f}_\mu(\boldsymbol{x}_t)\right\|^2 + L\eta^2\mathbb{E}\left\|\hat{G}_\mu\right\|^2. \tag{30}$$

Then, we sum up all (30) for all $t = 0, ..., T - 1$ and establish

$$\eta(1 - \frac{1}{2\epsilon})\sum_{t=0}^{T-1}\mathbb{E}\|\nabla f_\mu(\boldsymbol{x}_t)\|^2$$

$$\leq f_\mu(\boldsymbol{x}_0) - f_\mu(\boldsymbol{x}_T) + \frac{\eta\epsilon}{2}\sum_{t=0}^{T-1}\left\|\nabla f_\mu(\boldsymbol{x}_t) - \nabla\hat{f}_\mu(\boldsymbol{x}_t)\right\|^2 + L\eta^2\sum_{t=0}^{T-1}\mathbb{E}\left\|\hat{G}_\mu\right\|^2 \tag{31}$$

$$\leq f_\mu(\boldsymbol{x}_0) - f_\mu(\boldsymbol{x}^*) + \frac{\eta\epsilon}{2}\underbrace{\sum_{t=0}^{T-1}\left\|\nabla f_\mu(\boldsymbol{x}_t) - \nabla\hat{f}_\mu(\boldsymbol{x}_t)\right\|^2}_{A_1} + \underbrace{L\eta^2\sum_{t=0}^{T-1}\mathbb{E}\left\|\hat{G}_\mu\right\|^2}_{A_2}, \tag{32}$$

where the last inequality follows from $f_\mu(\boldsymbol{x}^*) \leq f_\mu(\boldsymbol{x}_T)$.

Next, we process $A_1$ as follows. We apply (15) on the first term and the third term of (33). For the second term of (33), we obtain it by Lemma 4.

Now, we deal with $A_1$:

$$A_1 = \left\|\nabla f_\mu(\boldsymbol{x}_t) - \nabla\hat{f}_\mu(\boldsymbol{x}_t)\right\|^2$$

$$= \left\| \nabla f_\mu(\boldsymbol{x}_t) - \nabla f(\boldsymbol{x}_t) + \nabla f(\boldsymbol{x}_t) - \nabla \hat{f}(\boldsymbol{x}_t) + \nabla \hat{f}(\boldsymbol{x}_t) - \nabla \hat{f}_\mu(\boldsymbol{x}_t) \right\|^2$$

$$\leq 3 \left\| \nabla f_\mu(\boldsymbol{x}_t) - \nabla f(\boldsymbol{x}_t) \right\|^2 + 3 \left\| \nabla f(\boldsymbol{x}_t) - \nabla \hat{f}(\boldsymbol{x}_t) \right\|^2 + 3 \left\| \nabla \hat{f}(\boldsymbol{x}_t) - \nabla \hat{f}_\mu(\boldsymbol{x}_t) \right\|^2 \quad (33)$$

$$\leq \frac{3}{4}\mu^2 L^2(d+3)^3 + 3(1+C_{S_1}^2)L_2^2\kappa^2 L_{S_1}^2 + \frac{3}{4}\mu^2 L^2(d+3)^3 \quad (34)$$

$$= \frac{3}{2}\mu^2 L^2(d+3)^3 + 3(1+C_{S_1}^2)L_2^2\kappa^2 L_{S_1}^2, \quad (35)$$

where we obtain (34) by Lemma 4.

Then, we start to deal with $A_2$.

$$A_2 = L\eta^2 \sum_{t=0}^{T-1} \mathbb{E} \left\| \hat{G}_\mu \right\|^2$$

$$\leq L\eta^2 \sum_{t=1}^{T} \left( \frac{\mu^2}{2} L^2(d+6)^3 + 2(d+4)\mathbb{E} \left\| \hat{G} \right\|^2 \right)$$

$$\leq L\eta^2 \sum_{t=0}^{T-1} \left( \frac{\mu^2}{2} L^2(d+6)^3 + 2(d+4) \left( \mathbb{E} \left\| \nabla \hat{f}(\boldsymbol{x}_t) \right\|^2 + \sigma^2 \right) \right)$$

$$= L\eta^2 \sum_{t=0}^{T-1} \left( \frac{\mu^2}{2} L^2(d+6)^3 + 2(d+4) \left( \mathbb{E} \left\| \nabla \hat{f}(\boldsymbol{x}_t) - \nabla f(\boldsymbol{x}_t) + \nabla f(\boldsymbol{x}_t) \right\|^2 + \sigma^2 \right) \right)$$

$$\leq L\eta^2 \sum_{t=0}^{T-1} \left( \frac{\mu^2}{2} L^2(d+6)^3 + 2(d+4) \left( 2\mathbb{E} \left\| \nabla \hat{f}(\boldsymbol{x}_t) - \nabla f(\boldsymbol{x}_t) \right\|^2 + 2\mathbb{E} \left\| \nabla f(\boldsymbol{x}_t) \right\|^2 + \sigma^2 \right) \right)$$

$$\quad (36)$$

$$\leq L\eta^2 \sum_{t=0}^{T-1} \left( \frac{\mu^2}{2} L^2(d+6)^3 + 2(d+4) \left( 2(1+C_{S_1}^2)L_2^2\kappa^2 L_{S_1}^2 + \sigma^2 \right) + 4(d+4)\mathbb{E} \left\| \nabla f(\boldsymbol{x}_t) \right\|^2 \right)$$

$$= L\eta^2 \sum_{t=0}^{T-1} \left( \frac{\mu^2}{2} L^2(d+6)^3 + 2(d+4) \left( 2(1+C_{S_1}^2)L_2^2\kappa^2 L_{S_1}^2 + \sigma^2 \right) \right) + 4(d+4)L\eta^2 \sum_{t=0}^{T-1} \mathbb{E} \left\| \nabla f(\boldsymbol{x}_t) \right\|^2,$$

$$\quad (37)$$

where we apply Lemma 4 on the term $\left\| \nabla \hat{f}(\boldsymbol{x}_t) - \nabla f(\boldsymbol{x}_t) \right\|^2$ of (36).

Putting all pieces above together, we establish:

$$(\eta - \frac{\eta}{2\epsilon}) \sum_{t=0}^{T-1} \mathbb{E} \left\| \nabla f_\mu(\boldsymbol{x}_t) \right\|^2 \leq f_\mu(\boldsymbol{x}_0) - f_\mu(\boldsymbol{x}^*) + \frac{\eta\epsilon}{2} \sum_{t=0}^{T-1} \left( \frac{3}{2}\mu^2 L^2(d+3)^3 + 3(1+C_{S_1}^2)L_2^2\kappa^2 L_{S_1}^2 \right)$$

$$+ L\eta^2 \sum_{t=0}^{T-1} \left( \frac{\mu^2}{2} L^2(d+6)^3 + 2(d+4) \left( 2(1+C_{S_1}^2)L_2^2\kappa^2 L_{S_1}^2 + \sigma^2 \right) \right)$$

$$+ 4(d+4)L\eta^2 \sum_{t=0}^{T-1} \mathbb{E} \left\| \nabla f(\boldsymbol{x}_t) \right\|^2 \quad (38)$$

By using (19) and re-arranging (38), we get

$$\left( \frac{\eta}{2}(1 - \frac{1}{2\epsilon}) - 4(d+4)L\eta^2 \right) \sum_{t=0}^{T-1} \mathbb{E} \left\| \nabla f(\boldsymbol{x}_t) \right\|^2$$

$$\leq f(\boldsymbol{x}_0) - f(\boldsymbol{x}^*) + \mu^2 Ld + \frac{\eta\epsilon}{2} \sum_{t=0}^{T-1} \left( \frac{3}{2}\mu^2 L^2(d+3)^3 + 3(1+C_{S_1}^2)L_2^2\kappa^2 L_{S_1}^2 \right)$$

$$+ L\eta^2 \sum_{t=0}^{T-1} \left( \frac{\mu^2}{2} L^2 (d+6)^3 + 2(d+4)\Big( 2(1+C_{S_1}^2)L_2^2\kappa^2 L_{S_1}^2 + \sigma^2 \Big) \right)$$

$$+ \eta(1 - \frac{1}{2\epsilon}) \sum_{t=0}^{T-1} \frac{\mu^2}{4} L^2 (d+3)^3 \tag{39}$$

We select $\epsilon = 1$ and further simplify

$$\left( \eta - 16(d+4)L\eta^2 \right) \sum_{t=0}^{T-1} \mathbb{E} \|\nabla f(\boldsymbol{x}_t)\|^2 \leq 4\big(f(\boldsymbol{x}_0) - f(\boldsymbol{x}^*)\big) + 4\mu^2 Ld + \eta\big(3\mu^2 L^2 (d+3)^3$$

$$+ L\eta^2 T \left( 2\mu^2 L^2 (d+6)^3 + 8(d+4)\Big( 2(1+C_{S_1}^2)L_2^2\kappa^2 L_{S_1}^2 + \sigma^2 \Big) \right)$$

$$+ 6(1+C_{S_1}^2)L_2^2\kappa^2 L_{S_1}^2)T + \frac{1}{2}\eta\mu^2 L^2 (d+3)^3 T \tag{40}$$

We assume that $\frac{1}{2}\eta \leq \big(\eta - 16(d+4)L\eta^2\big)$ and then get $\eta \leq \frac{1}{32(d+4)L}$. Then, we can further simplify the inequality above.

$$\frac{1}{T} \sum_{t=0}^{T-1} \mathbb{E} \|\nabla f(\boldsymbol{x}_t)\|^2 \leq \frac{8\big(f(\boldsymbol{x}_0) - f(\boldsymbol{x}^*)\big)}{T\eta} + \frac{8\mu^2 Ln}{T\eta} + 7\mu^2 L^2 (d+3)^3 + 12(1+C_{S_1}^2)L_2^2\kappa^2 L_{S_1}^2$$

$$+ 4L\eta \left( \mu^2 L^2 (d+6)^3 + 4(d+4)\Big( 2(1+C_{S_1}^2)L_2^2\kappa^2 L_{S_1}^2 + \sigma^2 \Big) \right) \tag{41}$$

Assuming that $\eta = \mathcal{O}(\frac{1}{\sqrt{Td}})$, then we obtain

$$\frac{1}{T} \sum_{t=0}^{T-1} \mathbb{E} \|\nabla f(\boldsymbol{x}_t)\|^2 = \mathcal{O}\left( \frac{D\sqrt{d}}{\sqrt{T}} \right) + \mathcal{O}\left( \frac{\mu^2 d^{\frac{3}{2}}}{\sqrt{T}} \right) + \mathcal{O}\Big(\mu^2 (d+3)^3\Big) + \mathcal{O}\Big((1+C_{S_1}^2)L_2^2\kappa^2 L_{S_1}^2\Big)$$

$$+ \mathcal{O}\left( \frac{\mu^2 (d+6)^3}{\sqrt{Td}} \right) + \mathcal{O}\left( \frac{(d+4)(1+C_{S_1}^2)L_2^2\kappa^2 L_{S_1}^2}{\sqrt{Td}} \right) + \mathcal{O}\left( \frac{(d+4)\sigma^2}{\sqrt{Td}} \right), \tag{42}$$

where $D = f(\boldsymbol{x}_0) - f(\boldsymbol{x}^*)$.

Further, we set $\mu \leq \frac{1}{(d+6)\sqrt{T}}$, then we can get

$$\frac{1}{T} \sum_{t=0}^{T-1} \mathbb{E} \|\nabla f(\boldsymbol{x}_t)\|^2 = \mathcal{O}\left( \frac{D\sqrt{d}}{\sqrt{T}} \right) + \mathcal{O}\left( \frac{1}{T\sqrt{d}} \right) + \mathcal{O}\left( \frac{d}{T} \right) + \mathcal{O}\Big((1+C_{S_1}^2)L_2^2\kappa^2 L_{S_1}^2\Big)$$

$$+ \mathcal{O}\left( \frac{\sqrt{d}}{T^{\frac{3}{2}}} \right) + \mathcal{O}\left( \frac{\sqrt{d}}{\sqrt{T}}(1+C_{S_1}^2)L_2^2\kappa^2 L_{S_1}^2 \right) + \mathcal{O}\left( \frac{\sqrt{d}}{\sqrt{T}}\sigma^2 \right)$$

Further, we simplify it, then we can get

$$\frac{1}{T} \sum_{t=0}^{T-1} \mathbb{E} \|\nabla f(\boldsymbol{x}_t)\|^2 = \mathcal{O}\left( \frac{D\sqrt{d}}{\sqrt{T}} \right) + \mathcal{O}\left( \frac{d}{T} \right) + \mathcal{O}\left( (1+C_{S_1}^2)L_2^2\kappa^2 L_{S_1}^2\Big( \frac{\sqrt{d}}{\sqrt{T}} + 1 \Big) \right) + \mathcal{O}\left( \frac{\sqrt{d}}{\sqrt{T}}\sigma^2 \right)$$

$$\blacksquare$$

### D.3 PROOF OF THEOREM 2

**Assumption 6 (Lipschitz Gradients)** *For multi-node cases ($M > 2$), we suppose that*

- *$f(\cdot)$ has L-Lipschitz gradient,*

- *$f \circ S_M \circ \cdots S_{i+1}$ has a $L_{f \circ S_M \circ \cdots S_{i+1}}$-Lipschitz gradient, and its gradient is bounded by $C_{f \circ S_M \circ \cdots S_{i+1}}$ for all $i = 1, ..., M-1$,*

- *The submodel $S_i$ is $L_{S_i}$-Lipschitz, and its gradient is bounded by $C_{S_i}$, for all $i = 1, ..., M$.*

**Lemma 5 (Distance between Gradients of Uncompressed and Compressed Activations, M>2)**
*For Alg. 3, the difference between gradients of uncompressed and compressed activations can be bounded by a constant term as follows:*

$$\left\|\nabla f(\boldsymbol{x}_t) - \nabla \hat{f}(\boldsymbol{x}_t)\right\|^2 \le 2M\kappa^2 L_S^2 \Psi$$

*Proof of lemma 5:*

For simplicity in the following proof, we denote that

$$\bar{S}_i = S_i(\cdots(S_2(S_1(\xi, \boldsymbol{x}_t^{(1)}); \boldsymbol{x}_t^{(2)}); \cdots); \boldsymbol{x}_t^{(i)}) \text{ and } \bar{m}_i = S_i(\cdots(\mathcal{C}(S_1(\xi; \boldsymbol{x}_t^{(1)})); \cdots); \boldsymbol{x}_t^{(i)}).$$

Then, we bound the gradient difference as follows:

$$\left\|\nabla f(\boldsymbol{x}_t) - \nabla \hat{f}(\boldsymbol{x}_t)\right\|^2$$

$$= \left\|\nabla_{\boldsymbol{x}^{(1)}} f(\boldsymbol{x}_t) - \nabla_{\boldsymbol{x}^{(1)}} \hat{f}(\boldsymbol{x}_t)\right\|^2 + \cdots + \left\|\nabla_{\boldsymbol{x}^{(M)}} f(\boldsymbol{x}_t) - \nabla_{\boldsymbol{x}^{(M)}} \hat{f}(\boldsymbol{x}_t)\right\|^2$$

$$= \sum_{i=1}^M \left\| \nabla_{S_i}(F \circ S_M \circ \cdots \circ S_{i+1})\big|_{(\bar{S}_i, \cdots, \boldsymbol{x}_t^{(M)})} \cdot \nabla_{\boldsymbol{x}^{(2)}} S_i \big|_{(\bar{S}_{i-1}, \boldsymbol{x}_t^{(i)})} \right.$$

$$\left. - \nabla_{S_i}(F \circ S_M \circ \cdots \circ S_{i+1})\big|_{(\bar{m}_i, \boldsymbol{x}_t^{(i+1)}, \cdots, \boldsymbol{x}_t^{(M)})} \cdot \nabla_{\boldsymbol{x}^{(i)}} S_i \big|_{(\bar{m}_{i-1}, \boldsymbol{x}_t^{(i)})} \right\|^2$$

$$\le (1 + 2C_{S_{M-1}}^2) L_{f \circ S_M}^2 \kappa^2 L_{S_{M-1}}^2 + 2\kappa^2 \sum_{i=1}^{M-2} (C_{S_{M-2}}^2 L_{f \circ S_M \circ \cdots \circ S_{i+1}}^2 + C_{f \circ S_M \circ \cdots \circ S_{i+2}}^2 L_{S_{i+1}}^2) L_{S_i}^2$$

$$\le \underbrace{2M\kappa^2 L_S^2 \max\left\{ (1 + 2C_{S_{M-1}}^2) L_{f \circ S_M}^2 \kappa^2, \max_{i \in [M-2]}\{C_{S_{M-2}}^2 L_{f \circ S_M \circ \cdots \circ S_{i+1}}^2 + C_{f \circ S_M \circ \cdots \circ S_{i+2}}^2 L_{S_{i+1}}^2\}\right\}}_{\Psi},$$

where $L_S^2 = \max\{L_{S_1}^2, \cdots, L_{S_M}^2\}$. ∎

**Theorem 2 (Convergence of `SparQ` under Non-Convexity, *M>2*)** *For Alg. 3, under assumptions 1, 2, 3, 5 and 6, if the number of computing nodes $M > 2$ and learning rate $\eta \le 1/32(d+4)L$, then the sequence $\{\boldsymbol{x}_t\}$ generated by* `SparQ` *satisfies*

$$\frac{1}{T} \sum_{t=0}^{T-1} \mathbb{E}\|\nabla f(\boldsymbol{x}_t)\|^2 \le \frac{8\big(f(\boldsymbol{x}_0) - f(\boldsymbol{x}^*)\big)}{T\eta} + \frac{8\mu^2 Ln}{T\eta} + 7\mu^2 L^2 (d+3)^3 + 12M\kappa^2 L_S^2 \Psi$$

$$+ 4L\eta\Big(\mu^2 L^2 (d+6)^3 + 4(d+4)(4M\kappa^2 L_S^2 \Psi + \sigma^2)\Big),$$

*where the definitions of $L_S$ and $\Psi$ can be found in the following proof.*

*Proof of Theorem 2:*

Here, we only provide a rough proof because we use the same proof framework as the proof of Theorem 1. The key differences are using Lemma 5 in inequalities (33) and (36).

Hence, $A_1$ and $A_2$ will be modified as follows:

$$A_1 = \left\|\nabla f_\mu(\boldsymbol{x}_t) - \nabla \hat{f}_\mu(\boldsymbol{x}_t)\right\|^2$$

$$= \left\|\nabla f_\mu(\boldsymbol{x}_t) - \nabla f(\boldsymbol{x}_t) + \nabla f(\boldsymbol{x}_t) - \nabla \hat{f}(\boldsymbol{x}_t) + \nabla \hat{f}(\boldsymbol{x}_t) - \nabla \hat{f}_\mu(\boldsymbol{x}_t)\right\|^2$$

$$\le 3\|\nabla f_\mu(\boldsymbol{x}_t) - \nabla f(\boldsymbol{x}_t)\|^2 + 3\left\|\nabla f(\boldsymbol{x}_t) - \nabla \hat{f}(\boldsymbol{x}_t)\right\|^2 + 3\left\|\nabla \hat{f}(\boldsymbol{x}_t) - \nabla \hat{f}_\mu(\boldsymbol{x}_t)\right\|^2$$

$$\le \frac{3}{4}\mu^2 L^2 (d+3)^3 + 6M\kappa^2 L_S^2 \Psi + \frac{3}{4}\mu^2 L^2 (d+3)^3 \tag{43}$$

$$= \frac{3}{2}\mu^2 L^2 (d+3)^3 + 6M\kappa^2 L_S^2 \Psi,$$

where we obtain (43) by Lemma 5.

$$
A_2 = L\eta^2 \sum_{t=0}^{T-1} \mathbb{E} \left\| \hat{G}_\mu \right\|^2
$$

$$
\leq L\eta^2 \sum_{t=1}^{T} \left( \frac{\mu^2}{2} L^2 (d+6)^3 + 2(d+4)\mathbb{E} \left\| \hat{G} \right\|^2 \right)
$$

$$
\leq L\eta^2 \sum_{t=0}^{T-1} \left( \frac{\mu^2}{2} L^2 (d+6)^3 + 2(d+4) \left( \mathbb{E} \left\| \nabla \hat{f}(\boldsymbol{x}_t) \right\|^2 + \sigma^2 \right) \right)
$$

$$
= L\eta^2 \sum_{t=0}^{T-1} \left( \frac{\mu^2}{2} L^2 (d+6)^3 + 2(d+4) \left( \mathbb{E} \left\| \nabla \hat{f}(\boldsymbol{x}_t) - \nabla f(\boldsymbol{x}_t) + \nabla f(\boldsymbol{x}_t) \right\|^2 + \sigma^2 \right) \right)
$$

$$
\leq L\eta^2 \sum_{t=0}^{T-1} \left( \frac{\mu^2}{2} L^2 (d+6)^3 + 2(d+4) \left( 2\mathbb{E} \left\| \nabla \hat{f}(\boldsymbol{x}_t) - \nabla f(\boldsymbol{x}_t) \right\|^2 + 2\mathbb{E} \left\| \nabla f(\boldsymbol{x}_t) \right\|^2 + \sigma^2 \right) \right)
$$

$$
\leq L\eta^2 \sum_{t=0}^{T-1} \left( \frac{\mu^2}{2} L^2 (d+6)^3 + 2(d+4) \left( 4M\kappa^2 L_S^2 \Psi + \sigma^2 \right) + 4(d+4)\mathbb{E} \left\| \nabla f(\boldsymbol{x}_t) \right\|^2 \right)
$$

$$
= L\eta^2 \sum_{t=0}^{T-1} \left( \frac{\mu^2}{2} L^2 (d+6)^3 + 2(d+4)(4M\kappa^2 L_S^2 \Psi + \sigma^2) \right) + 4(d+4)L\eta^2 \sum_{t=0}^{T-1} \mathbb{E} \left\| \nabla f(\boldsymbol{x}_t) \right\|^2,
$$

Then, the subsequent steps are the same as the proof of Theorem 1, so we skip them and directly arrive at

$$
\left( \eta - 16(d+4)L\eta^2 \right) \sum_{t=0}^{T-1} \mathbb{E} \left\| \nabla f(\boldsymbol{x}_t) \right\|^2 \leq 4\big( f(\boldsymbol{x}_0) - f(\boldsymbol{x}^*) \big) + 4\mu^2 Ld + \eta \big( 3\mu^2 L^2 (d+3)^3
$$

$$
+ L\eta^2 T \left( 2\mu^2 L^2 (d+6)^3 + 8(d+4)(4M\kappa^2 L_S^2 \Psi + \sigma^2) \right) + 6M\kappa^2 \Psi \big) T + \frac{1}{2}\eta\mu^2 L^2 (d+3)^3 T
$$

We assume that $\frac{1}{2}\eta \leq \left( \eta - 16(d+4)L\eta^2 \right)$ and then get $\eta \leq \frac{1}{32(d+4)L}$. Then, we can further simplify the inequality above and obtain

$$
\frac{1}{T} \sum_{t=0}^{T-1} \mathbb{E} \left\| \nabla f(\boldsymbol{x}_t) \right\|^2 \leq \frac{8\big( f(\boldsymbol{x}_0) - f(\boldsymbol{x}^*) \big)}{T\eta} + \frac{8\mu^2 Ln}{T\eta} + 7\mu^2 L^2 (d+3)^3 + 12M\kappa^2 L_S^2 \Psi
$$

$$
+ 4L\eta \Big( \mu^2 L^2 (d+6)^3 + 4(d+4)(4M\kappa^2 L_S^2 \Psi + \sigma^2) \Big)
$$

$\blacksquare$

