# OpenReview forum: "Breaking Memory and Communication Barriers in Model-Parallel Fine-Tuning of Large Language Models"
_ICLR.cc/2026/Conference — Submitted to ICLR 2026_

### Official Review · Reviewer_LgFa · 2025-10-27

**Soundness:** 2
**Presentation:** 2
**Contribution:** 2
**Rating:** 2
**Confidence:** 4

**Summary:**

This paper tried zero order (ZO) methods on multi-nodes. Then do activation quantization to reduce communication cost.

**Strengths:**

Good quantitive analysis in experiments.

Nice and informative figure 2

**Weaknesses:**

1. The authors may not have a deep understanding of model parallelism frameworks themselves. For example, in Page1 L41-42, ZeRO belongs to DeepSpeed, and ZeRO itself is not a open-source "framework"

2. these kind of activation quantization has be well-studied previously, such as ZeRO++[1], LLM-FP4[2] just to name a few. With (FO) or without (ZO) backward does not change anything on the forward propagation of activations. Thus the major novelty of this paper is very similar to existed work.

3. The reason for ZO not being widely used is due to its instability and may lead to worse model accuracy. The authors do not propose anything to fix this major issue of ZO.

4. the experiments use vey old models (e.g., llama-1b, gpt2), which can not be representatives of latest LLM results.

5. Indeed, split layer like pipeline parallelism, there is no any training logic difference compared with training whole model on single device. Thus making the problem itself (i.e. quote page 1 L52-53: "the potential of applying ZO optimization within a MP framework for fine-tuning LLMs remains unexplored.") not very fundamental and meaningful.

6. some equations are over-decorated. For example, I don't see any one represent such simple Pipeline parallelism idea where one device's layer output serve as next device layer input, to such a complicated equation on L127 of page 3.

[1] ZeRO++: Extremely Efficient Collective Communication for Large Model Training, ICLR 2024

[2] LLM-FP4: 4-Bit Floating-Point Quantized Transformers, EMNLP 2023

**Questions:**

N/A

---

> ### Author Response · Authors · 2025-11-30
>
> > ### W1: About ZeRO.
>
> **Our response:** Thank you for the clarification. We agree that our wording in L41–42 was imprecise: ZeRO is an optimization technique within the DeepSpeed framework rather than an independent open-source framework. We will revise the text accordingly. This correction does not affect our technical claims or any part of the SparQ methodology.
>
> ---
>
> > ### W2: The major novelty of this paper
>
> **Our response:** Thank you for the comment. We believe that there is a misunderstanding about our novelty and contributions. Our key contribution/novelty is the sparsity-guided split strategy, which leverages quantization-induced activation sparsity to determine where to partition the model in a model-parallel ZO setting. Thus, SparQ introduces a new use of activation quantization not as the end goal, but as a mechanism enabling the sparsity-guided partitioning strategy that substantially reduces communication in ZO-based model-parallel fine-tuning. We will clarify this distinction more clearly in the revision.
>
> ---
>
> > ### W3: About ZO's instability.
>
> **Our response:** Thank you for the comment. We agree that ZO methods may exhibit less stability than FO methods in certain scenarios, but addressing ZO’s inherent optimization stability is not the objective of this work. The goal of this work is to target the memory and communication barriers in model-parallel frameworks. Moreover, a common way to stablize ZO methods is to increase the number of perturbations, which has been validated in prior works [R1, R2]. We did not focus on this just because this is not our goal and contributions.
>
> [R1] Liu, S., Chen, P. Y., Kailkhura, B., Zhang, G., Hero III, A. O., & Varshney, P. K. (2020). A primer on zeroth-order optimization in signal processing and machine learning: Principals, recent advances, and applications. IEEE Signal Processing Magazine, 37(5), 43-54.
>
> [R2] Li, Z., Ying, B., Liu, Z., Dong, C., & Yang, H. Achieving Dimension-Free Communication in Federated Learning via Zeroth-Order Optimization. In The Thirteenth International Conference on Learning Representations.
>
> ---
>
> > ### W4: About using old models.
>
> **Our response:** Thank you for the comment. Our goal in this work is to evaluate the memory and communication behavior of model-parallel ZO fine-tuning, which doesn not depend on whether the model is the latest LLM. Models such as GPT-2, OPT, and Llama-1B already exhibit the same activation patterns, quantization behavior, and communication bottlenecks that SparQ targets, and they are widely used in recent ZO fine-tuning literature to enable controlled comparisons. We agree that including larger and newer models could further strengthen the empirical section, and we will explore incorporating them in a future revision; however, the structural efficiency gains demonstrated by SparQ do not depend on the specific model vintage.
>
> ---
>
> > ### W5: The Meaningfulness of ZO in model parallelism
>
> **Our response:** Thank you for the comment. While we agree that pipeline-style splitting does not change the training logic for FO methods, the situation is fundamentally different for ZO optimization. Unlike FO training, ZO requires multiple forward passes per iteration and has no backward graph, which makes inter-partition activation transmission the dominant cost in a model-parallel environment. As a result, the communication pattern, memory behavior, and partitioning considerations for ZO differ substantially from those in FO pipeline parallelism. This is precisely why existing MP frameworks and FO-oriented activation compression techniques do not directly apply, and why the potential of ZO within MP settings has remained unexplored. SparQ's sparsity-guided split strategy is designed specifically to address these ZO-specific system bottlenecks.
>
> ---
>
> > ### W6: The Formalization of the MP Computation
>
> **Our response:** Thank you for the comment. The compositional formulation we use is standard in optimization papers, including works such as AQ-SGD, because it provides the mathematical structure required for analyzing perturbations, communication, and convergence in the model-parallel setting. Our goal was to follow this convention rather than to complicate the idea.

---

### Official Review · Reviewer_zAG3 · 2025-11-01

**Soundness:** 2
**Presentation:** 3
**Contribution:** 2
**Rating:** 4
**Confidence:** 4

**Summary:**

The paper proposes a zeroth-order (ZO) model-parallel framework SparQ that drastically reduces the communication and memory overhead of LLM fine-tuning. SparQ uses gradient-free ZO optimization, paired with quantization-induced activation sparsity to minimize data transfer between partitions. By placing model split layers where activations become sparse after quantization, only a small number of nonzero activations are transmitted in a sparse format, yielding large efficiency gains, while maintaining comparable fine-tuning accuracy.

**Strengths:**

1.	The paper presents a clear and intuitive motivation, and the overall logic of the work is easy to follow.
2.	By demonstrating consistent results across models of different sizes, the paper provides strong empirical support for the proposed approach and makes the study more convincing overall.
3.	The authors support their system design with a non-convex convergence guarantee matching centralized ZO methods, reinforcing technical soundness beyond empirical evidence.

**Weaknesses:**

1.	SparQ’s effectiveness heavily depends on activation sparsity after quantization, which holds for ReLU/GELU-based transformers but becomes unstable in layers with normalization, attention, or mixed activations. This makes the “sparsity-guided split” design less generalizable across architectures and limits its scalability to more diverse LLMs.
2.	The paper primarily evaluates on GLUE and SuperGLUE benchmarks, which are relatively simple for LLM fine-tuning under both first- and zeroth-order methods. Including results on more challenging tasks would further strengthen the work. This is a suggestion rather than a weakness.
3.	The authors conduct memory profiling without using gradient checkpointing, which is a common technique to reduce activation memory during backpropagation. This likely overestimates the memory footprint of FO-SGD. Since the main advantage of SparQ lies in its lower memory usage compared to FO-SGD, this comparison becomes less compelling once gradient checkpointing is enabled.
4.	Although the authors claim that their framework can also be easily extended to scenarios with multiple computing nodes (M > 2). However, I haven't seen any experimental results when M>2, so the results only when M=2 are not convincing enough for me.

**Questions:**

1.	How would results look like when if running FO-SGD with gradient checkpointing?
2.	The 4-Bit Quantized Activations Comparison (Fig4) is very interesting, and I'm curious about how the author defines trainable and untrainable. If you define it by accuracy, I'd like to know what the accuracy is when it's untrainable compared to when it's trainable.

---

> ### Author Response · Authors · 2025-11-30
>
> > ### W1: The generalizability of SparQ’s sparsity-guided split strategy.
>
> **Our response:** Thank you very much for raising this concern. We would like to clarify that SparQ does not rely on a specific activation type (e.g., ReLU) for its effectiveness. As shown in our experiments, quantization consistently induces high activation sparsity across ReLU, GELU, and SwiGLU models (OPT, GPT-2, Llama-1B), even though their unquantized activations have very different sparsity patterns. This makes the sparsity-guided split strategy stable and architecture-agnostic in practice. Moreover, our splitting is performed immediately after activation functions, which occur after normalization and attention computations. Therefore, potential instability in those modules does not affect the sparsity pattern of the split layer.
>
> We agree that extending SparQ to architectures with substantially different activation structures is an interesting direction, but our results demonstrate that the proposed design already generalizes well across diverse and widely used LLM families.
>
> ---
>
> > ### W2: About more challenging tasks.
>
> **Our response:** Thank you for this insightful suggestion. We would like to clarify that task difficulty has minimal impact on the core quantities SparQ aims to improve: memory consumption and communication cost. These costs are determined primarily by models, activation sparsity, and optimizer design, rather than by benchmark complexity. Therefore, following prior ZO fine-tuning works, we adopt GLUE/SuperGLUE as controlled benchmarks to isolate and compare SparQ’s efficiency gains under identical settings. We agree that adding more challenging tasks is valuable and consider it a promising extension for future work.
>
> ---
>
> > ### W3&Q1: About checkpointing.
>
> **Our response:** Thank you for the insightful comment. We agree that gradient checkpointing is a useful technique to reduce forward activation memory for FO-SGD. However, our main memory advantage does not come from reducing forward activations, but from eliminating the entire backward propagation, which dominate the memory footprint in LLMs. As shown in prior works and confirmed in our experiments, even with checkpointing, FO-SGD still requires storing high-dimensional gradients and optimizer states, resulting in memory usage several times larger than the model size. In contrast, SparQ's memory cost is approximately equal to one forward pass and thus remains unaffected by such techniques. Thus, while checkpointing can slightly reduce FO-SGD's activation memory, it does not change the fundamental memory gap between FO methods and gradient-free ZO methods. Incorporating checkpointing would not affect the validity of our conclusions.
>
> ---
>
> > ### W4: About $M>2$ results.
>
> **Our response:** Thank you for your question. Our theoretical formulation and algorithm (Appendix C.3) explicitly support $M>2$ by recursively applying the same sparsity-guided split and scalar-only backward communication. The communication and memory behavior simply repeat across additional partitions and do not introduce new challenges specific to SparQ, which is why $M=2$ is commonly reported in prior model-parallel works. Moreover, we agree that including $M>2$ empirical results would further strengthen the paper, and we will add these experiments in the next revision.
>
> ---
>
> > ### Q2: About trainable and untrainable in Fig.4.
>
> **Our response:** Thank you so much for your interest in our Fig.4. In Fig. 4, trainable vs. untrainable is defined by whether the model can obtain meaningful test accuracy. When the split is placed after activation functions, the model converges normally and reaches the accuracy shown in Table 1-2. In contrast, when splitting between blocks, the 4-bit quantized activations become severely distorted, and the model's test accuracy stays close to zero throughout training, indicating that it is untrainable. We will clarify this definition and report these accuracy values in the revision.

---

### Official Review · Reviewer_8CVv · 2025-11-01

**Soundness:** 3
**Presentation:** 3
**Contribution:** 3
**Rating:** 6
**Confidence:** 2

**Summary:**

This paper proposes SparQ, a ZO MP framework with Split layer allocation informed by Quantization-induced activation sparsity. The paper proves that SparQ achieves a sublinear convergence rate in non-convex settings and empircally shows that it reduces GPU and communication overheads.

**Strengths:**

1. The paper proves that SparQ across multiple nodes achieves a sublinear convergence rate of for non-convex functions.
2. The evaluation provides test accuracies of their method accross models and datasets.

**Weaknesses:**

1. The experiments do not appear to include real-world distributed training across multiple machines, making it difficult to reflect the end-to-end training time advantages of SparQ in real-world scenarios. Meanwhile, memory cost and communication cost can be easily calculate without experiments in real-world hardware settings.

**Questions:**

What hardware environments did you use in your experiments?

What is the expected improvement in end-to-end convergence speed?

---

> ### Author Response · Authors · 2025-11-30
>
> > ### W1&Q2: About real-world distributed learning settings; End-to-end convergence speedup.
>
> **Our response:** We sincerely thank the reviewer for the thoughtful comments. We would like to clarify that our work is positioned as an optimization/algorithmic contribution, rather than a system paper. Our SparQ focuses on the algorithmic reduction of memory and communication complexity in model-parallel fine-tuning (Secs. 3–4), and these quantities can be precisely characterized analytically and validated through controlled experiments, without requiring full multi-machine deployment.
>
> End-to-end wall-clock measurements/convergence speedup depend heavily on system engineering choices (network stack, scheduling, hardware topology, etc.), which are outside the scope of our methodological contribution and are not necessary to demonstrate the benefits of the proposed optimizer. Our experiments have shown consistent memory reduction and significant communication savings, confirming that SparQ’s algorithmic advantages translate into practice.
>
> ---
>
> > ### Q1: Hardware environments.
>
> **Our response:** Thank you for your question. All experiments were conducted on NVIDIA A100 GPUs.

---

### Official Review · Reviewer_paKH · 2025-11-03

**Soundness:** 3
**Presentation:** 3
**Contribution:** 2
**Rating:** 4
**Confidence:** 4

**Summary:**

This paper introduces SparQ, a framework that combines zeroth-order (ZO) optimization, quantization-induced activation sparsity, and sparsity-aware split-layer allocation to reduce memory and communication overhead during model-parallel (MP) fine-tuning of large language models (LLMs). The key observation is that post-quantization activations (after ReLU, GELU, SwiGLU) become highly sparse, enabling sparse transmission between partitions.

**Strengths:**

1. Clearly motivated by real bottlenecks in model-parallel fine-tuning.

2. Integrates ZO optimization and activation quantization in a coherent framework.

3. Provides a theoretical convergence analysis under compression.

4. Demonstrates substantial GPU memory and communication savings with minor performance loss.

5. Ablation studies on quantization levels and split positions provide useful insights.

**Weaknesses:**

1. The evaluation lacks direct comparisons against strong existing baselines such as [1] or [2]. Since these methods are designed for similar purposes—reducing memory and communication in fine-tuning—the absence of head-to-head experiments makes it difficult to quantify SparQ’s real advantage.

[1] Zeroth-Order Fine-Tuning of LLMs with Extreme Sparsity

[2] ZO2: Scalable Zeroth-Order Fine-Tuning for Extremely Large Language Models with Limited GPU Memory

2. The largest model tested is 6.7B parameters (OPT-6.7B). In practice, such a size can often fit on a single GPU or two GPUs with ZeRO-2/ZeRO-3, and rarely requires model parallelism. If the proposed method is specifically targeting MP fine-tuning, stronger evidence on larger-scale (>13B or multi-node) models is expected. The current scale undermines the relevance of MP in this context.

3. Although communication and memory reductions are clearly reported, there are no measurements of actual training speed, wall-clock time, or GPU utilization. ZO optimization introduces multiple forward passes per step, which may offset the savings. Without runtime comparisons, the claimed “efficiency” remains uncertain.

4. The convergence proof assumes unbiased and bounded compression noise, yet sparse quantization is inherently biased. The paper does not measure or analyze the effect of this bias on convergence or final accuracy.

5. The evaluation does not cover multi-split (M>2) or multi-node settings, and no communication profiling under realistic hardware interconnects (e.g., NVLink, RDMA, Ethernet) is presented. Thus, the practical scalability and deployment feasibility are unclear.

6. The default configuration (4-bit quantization) is fixed throughout, but the trade-off between bitwidth, accuracy, and communication volume is not fully explored. It remains unclear whether the observed gains hold under different precision levels or tasks.

7. The cost of sparse activation encoding/decoding and synchronization is omitted. These may introduce non-negligible overhead that counteracts communication savings. More system-level profiling (kernel time, communication overlap) would strengthen the paper’s claims.

**Questions:**

1. Can you report end-to-end runtime comparisons (e.g., total fine-tuning time vs. MeZO, ZO-Adam, or first-order methods)?

2. Does increasing P significantly affect the communication–accuracy trade-off?

3. How large is the bias introduced by the sparse quantizer, and does it affect convergence?

4. Can SparQ scale to multi-node / multi-partition (M > 2) configurations?

5. How does SparQ perform on larger models (> 70B) that truly require MP fine-tuning?

6. What is the computational cost of sparse activation reconstruction during forward/backward passes?

---

> ### Author Response · Authors · 2025-11-30
>
> > ### W1: Comparison with two existing baselines.
>
> **Our response:** Thank you for raising this point. We carefully examined the two referenced works:
>
> [1] focuses on sparsifying model parameters during ZO-based fine-tuning. It does not address communication in model-parallel training, which is the core bottleneck SparQ targets. Thus, although related conceptually, [1] is orthogonal to our objective and not a direct baseline.
>
> [2] aims to reduce data transfer between CPU and GPU. This problem setting is fundamentally different from SparQ, which addresses inter-node communication in model-parallel LLM fine-tuning.
>
> Given these differences, neither [1] nor [2] constitutes a directly comparable baseline for SparQ’s targeted problem: reducing memory and activation communication in model-parallel LLM fine-tuning. Our experiments therefore focus on the most relevant AQ-SGD, which operate under the same MP setting and optimize the same communication pathway. We will clarify this distinction in the revision.
>
> ---
>
> > ### W2&Q5: About experiments on larger models.
>
> **Our response:** Thank you for the helpful comment. We would like to clarify that SparQ’s core advantages, which are activation-sparsity-driven communication reduction and the removal of the backward propagation, are structural properties of the method and do not depend on the model size. In practice, models we used already exhibited the same activation dimensionality, sparsity behavior, and communication patterns as even larger LLMs, making them sufficient to show the effectiveness of SparQ.
>
> Moreover, we fully agree that evaluating larger models (e.g., >13B) may further strengthen the empirical evidence, and we will try our best to include such experiments in our future revision. However, the current scale already captures the key behaviors SparQ is designed to improve and is adequate for validating its effectiveness.
>
> ---
>
> > ### W3&Q1: Runtime measurements and practical efficiency; End-to-end runtime comparisons.
>
> **Our response:** Thank you for the comment. SparQ focuses on reducing the structural communication and memory bottlenecks of model-parallel ZO LLM fine-tuning, and these improvements are independent of specific runtime implementations or GPU scheduling. We agree that wall-clock speed and utilization measurements would further strengthen the practical perspective. ZO’s extra forward passes are inherent to the baseline method rather than introduced by SparQ, and our goal is to make these passes feasible under MP settings by removing backward memory and activation communication costs.
>
> ---
>
> > ### W4: About unbiased assumption.
>
> **Our response:** Thank you for pointing out this. We agree that sparse quantization is biased. We checked our proof, and we found that our proof does not rely on the unbiasedness property in fact, so the current assumption is unnecessarily strong. We will revise this assumption then.
>
> ---
>
> > ### W5&Q4: About multi-node settings and hardware-level profiling.
>
> **Our response:** Thank you for the comment. SparQ's communication and memory reductions stem from structural properties (i.e., the sparsity-guided split and scalar-only backward) which do not depend on specific interconnect hardware. While we agree that adding multi-split ($M>2$) and multi-node results under NVLink/RDMA/Ethernet would further strengthen the system side, the current $M=2$ experiments already capture the core scaling behavior in model-parallel ZO fine-tuning. We will try our best to include extra multi-node evaluations in the future revision.
>
> ---
>
> > ### W6: About trade-off.
>
> **Our response:** Thank you for the comment. As an optimization-focused work, our goal is to study how quantization-induced sparsity enables a more efficient partitioning strategy in model-parallel ZO training, rather than to benchmark all possible precision levels. We use 4-bit as a standard low-precision configuration to keep the analysis controlled, and our results show that the key effect, activation sparsity that drives communication reduction, stems from the structural behavior of the quantizer, not from this specific bitwidth. Moreover, we agree that exploring a wider range of bitwidth-accuracy-communication trade-offs would enhance the empirical section, and we will include additional precision configurations in the revision.
>
> ---
>
> > ### W7: Encoding/decoding overhead and system-level profiling.
>
> **Our response:** Thank you for the comment. The cost of sparse activation encoding, decoding, and synchronization is minimal because these are simple element-wise quantization and zero-fill operations whose complexity is negligible compared with a Transformer forward pass.

---

> ### Author Response · Authors · 2025-11-30
>
> > ### Q2: About the impact of increasing $P$ on communication–accuracy trade-off.
>
> **Our response:** Thank you for the question. Increasing $P$ leads to more forward passes per round, which linearly increases the communication cost of a single round. However, a larger $P$ also reduces the variance of the ZO gradient estimator, making convergence more stable and often reducing the total number of required rounds.
>
> ---
>
> > ### Q3: About bias introduced by the sparse quantizer.
>
> **Our response:** Thank you for the question. While sparse quantization does introduce bias, in practice we observe that the magnitude of this bias is small and does not hinder convergence: the models trained with SparQ match the accuracy of standard ZO-SGD across all tasks in Tables 1–2.
>
> ---
>
> > ### Q6: About the computational cost of sparse activation reconstruction.
>
> **Our response:** Thank you for your question. SparQ only transmits and reconstructs activations at the split boundary, and reconstruction is a simple element-wise dequantization followed by zero-filling, which is linear in the activation dimension and significantly cheaper than a Transformer layer’s forward computation. Since our work focuses on optimization- and communication-level efficiency rather than micro-kernel engineering, we did not separately benchmark this overhead, but in practice it is dominated by the standard forward pass and does not affect overall training efficiency.

---

### Meta-Review · Area_Chair_SgCY · 2026-01-06

**Summary:**

The reviewers broadly agree that the paper presents a well-motivated framework for reducing memory and communication costs in model-parallel fine-tuning via zeroth-order optimization and quantization-induced activation sparsity. However, several reviewers raised substantive concerns regarding the practical relevance and empirical validation of the approach. In particular, the evaluation is limited to relatively small models and simplified settings, lacks end-to-end runtime measurements, and does not convincingly demonstrate advantages in realistic multi-node or large-scale model-parallel scenarios. Questions about novelty relative to existing activation compression and model-parallel techniques, as well as the generality of the sparsity-guided split strategy, further weakened confidence in the overall contribution.

**Reviewer Concerns:**

The rebuttal satisfactorily addressed some technical clarifications, including the role of biased quantization in the convergence analysis, wording issues around prior systems, and the conceptual distinction between SparQ and certain zeroth-order baselines.

**Reviewer Scores:**

Reviewer scores would likely remain unchanged due to unresolved empirical concerns.

---

### Decision · Program_Chairs · 2026-01-26

Reject